# Beyond Single Models: Mitigating Multimodal Hallucinations via Adaptive Token Ensemble Decoding

## Abstract

Large Vision-Language Models (LVLMs) have recently achieved impressive results in multimodal tasks such as image captioning and visual question answering. However, they remain prone to **object hallucination**—generating descriptions of nonexistent or misidentified objects. Prior work has partially mitigated this via auxiliary training objectives or external modules, but challenges remain in terms of scalability, adaptability, and model independence. To address these limitations, we propose **Adaptive Token Ensemble Decoding (ATED)**, a training-free, token-level ensemble framework that mitigates hallucination by aggregating predictions from multiple LVLMs during inference. ATED dynamically computes uncertainty-based weights for each model, reflecting their reliability at each decoding step. It also integrates diverse decoding paths to improve contextual grounding and semantic consistency. Experiments on standard hallucination detection benchmarks demonstrate that ATED significantly outperforms state-of-the-art methods, reducing hallucination without compromising fluency or relevance. Our findings highlight the benefits of adaptive ensembling and point to a promising direction for improving LVLM robustness in high-stakes applications.

## 1 Introduction

In recent years, large language models (LLMs) have made significant breakthroughs in natural language processing (Touvron et al., 2023; Chiang et al., 2023; Achiam et al., 2023; Bai et al., 2023a) and have been increasingly extended to vision-language tasks, giving rise to large vision-language models (LVLMs) (Ye et al., 2023; Liu et al., 2023; Li et al., 2023a; 2024; Chen et al., 2024c;d; Bai et al., 2023b). These models have demonstrated strong capabilities in both understanding (Zhang et al., 2025; Lai et al., 2024) and generating (Geng et al., 2023) multimodal content.

However, LVLMs often suffer from the problem of *object hallucination*, where the model generates details or objects that do not exist in the image (Li et al., 2023d; Wang et al., 2023; Gunjal et al., 2024; Liu et al., 2024b), significantly limiting their reliability in high-stakes applications, such as autonomous driving, medical image analysis, and remote sensing, where factual correctness and visual grounding are critical.

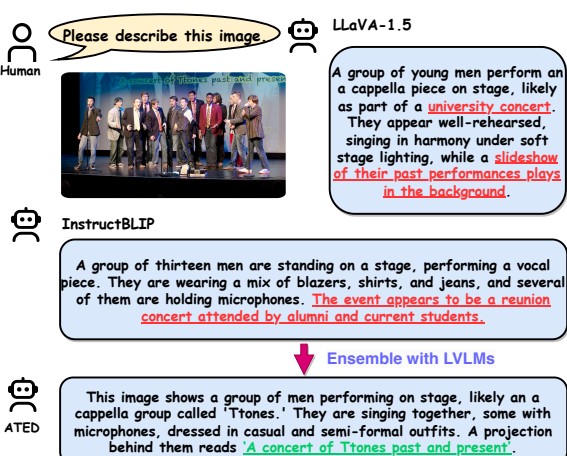

Figure 1: Comparison of image-description generation results from various LVLMs and our proposed *ATED* method. Red text indicates hallucination. Green text represents hallucination mitigating from *ATED*.

Early research on mitigating hallucinations primarily focused on enhancing data quality and training paradigms. Specifically, diverse instruction-tuning datasets and multi-task training approaches were introduced to reduce the models' tendency to hallucinate during generation (Li et al., 2023c; Liu et al., 2024a). Other methods adopted post-hoc strategies by implementing output-checking mechanisms to detect and correct hallucinated content (Yin et al., 2024; Zhou et al., 2024).

More recently, training-free approaches have emerged, including contrastive decoding methods, such as VCD (Leng et al., 2023) and ICD (Wang et al., 2024), retrospection-based decoding strategies (Huang et al., 2024) and token-level pruning techniques (Favero et al., 2024; Woo et al., 2024). However, challenges remain regarding cross-model and cross-task generalization. Furthermore, single-model strategies are inherently limited by the knowledge scope of the underlying model, restricting generalization and adaptability. As shown in Figure 1, existing LVLMs still exhibit notable levels of hallucination in image captioning tasks, highlighting the need for a more robust solution.

Ensemble learning Polikar (2012), which leverages the collective intelligence of multiple models, has proven highly effective at reducing errors and enhancing robustness in traditional classification and regression tasks Mienye & Sun (2022). More recently, it has been successfully extended to text generation tasks, particularly in LLMs, to enhance output accuracy and mitigate issues such as hallucination (Jiang et al., 2023; Wan et al., 2024). These advances suggest that ensemble and diversified inference strategies hold great promise for further improving the performance of generative models and enhancing their generalization and adaptability in complex tasks.

Inspired by these advances, this paper proposes a novel framework, **Adaptive Token Ensemble Decoding (ATED)**, which integrates ensemble-based decoding strategies into the autoregressive generation process in order to fully leverage the complementary strengths of different LVLMs. ATED is the first fine-grained, token-level ensemble method tailored for multimodal LVLMs. Without requiring any additional training, ATED enables parallel inference across multiple LVLMs and adaptively fuses their output logits at the token level through weighted aggregation. Specifically, we introduce an Uncertainty-Guided Weighting mechanism that quantifies each model's hallucination tendency at every decoding step based on output uncertainty, and employ a greedy optimization algorithm to minimize overall uncertainty and adaptively assign importance weights to each model. Moreover, ATED allows the adjustment of optimization thresholds and search space to flexibly balance performance and inference efficiency across various multimodal tasks. By aggregating outputs from multiple decoding paths, ATED not only substantially improves the factual accuracy and consistency of generated content, but also suppresses hallucinations and demonstrates strong adaptability and scalability.

Our main contributions are summarized as follows:

- We propose **ATED**, a training-free multimodal ensemble decoding method that mitigates hallucinations via fine-grained token-level fusion.

- We introduce an uncertainty-minimization weighting mechanism that dynamically assigns weights based on model confidence, improving the reliability of ensemble decoding.

- Extensive experiments show that ATED consistently outperforms existing methods across multiple multimodal benchmarks, achieving superior accuracy and robustness.

## 2 RELATED WORK

**Hallucination in LVLMs.** Hallucination was initially observed in LLMs, referring to generated content that deviates from factual knowledge or user intent (Jing et al., 2024; Liu et al., 2024b). Large vision-language models (LVLMs) (Bai et al., 2025; Zhang et al., 2023), which extend LLMs with visual inputs, also exhibit hallucinations—typically manifesting as mismatches between generated text and visual content. Existing studies categorize hallucinations in LVLMs into three main types: *object hallucination* (Biten et al., 2021; Li et al., 2023d; Rohrbach et al., 2019), *attribute hallucination*, and *relationship hallucination* (Wu et al., 2024; Zhou et al., 2024). Object hallucination refers to fabricated or omitted objects; attribute hallucination involves incorrect properties such as color or size; relationship hallucination describes inaccurate relations among objects. These errors may arise from visual misinterpretation, flawed reasoning, or overreliance on language priors.

**Hallucination Mitigation in LVLMs.** To address hallucination in LVLMs, researchers have proposed a variety of solutions, including improved instruction tuning(Jiang et al., 2024; Liu et al., 2024a; Yu et al., 2024a; Yue et al., 2024), reinforcement learning with human or AI feedback (Gunjal et al., 2024; Kim et al., 2024; Li et al., 2023b; Yu et al., 2024b), retrieval augmentation, and architectural improvements (Zhai et al., 2024). More recently, several training-free decoding strategies have been developed to suppress hallucinations in LVLMs. For example, conservative decoding methods that operate on both original and perturbed inputs (Leng et al., 2023; Wang et al., 2024; Chen et al., 2024b; Huo et al., 2025) aim to reduce overreliance on language priors. Approaches such as input distortion, which can be applied to either visual content or instructions, aim to amplify and subsequently identify hallucinations through contrastive decoding. In addition, token-level pruning and related methods (Favero et al., 2024; Woo et al., 2024) manipulate visual inputs to mitigate erroneous outputs.

**Summary.** Although these methods have shown effectiveness on certain benchmarks and tasks, challenges related to scalability and cross-domain generalization still persist. In contrast, our work explores a training-free ensemble approach that aims to fully leverage the complementary strengths of multiple models through output fusion, with the goal of improving applicability across diverse tasks and more effectively suppressing hallucinations.

## 3 METHODOLOGY

### 3.1 PRELIMINARIES OF LVLMS GENERATION

The generation mechanism of LVLMs can be deconstructed into three core modules: Vision Language Input, Model Forward Propagation, and Next Token Decoding.

**Vision Language Input.** LVLMs take both visual input $v$ and textual query input $q$. Specifically, the input image is first processed by a vision encoder (e.g., a pre-trained visual backbone), and the resulting features are then projected into the input space of the language decoder via a cross-modal interface module. Finally, this projected visual input $v$, combined with the corresponding textual query $q$, is fed into the language decoder for subsequent generation tasks.

**Model Forward Propagation.** Following the autoregressive generation paradigm, an LVLM parameterized by $\phi$, predicts the probability of the next token $x_t$ at time step $t$ based on the previously generated tokens, the input text, and the visual features, over the vocabulary set $\boldsymbol{O}$. This process can be formally expressed as:

$$p(x_t \mid v, q, x_{<t}) = \text{softmax}(\text{logit}_\phi(x_t \mid v, q, x_{<t})), \tag{1}$$

where $x_t \in \boldsymbol{O}$ denotes the token at time step $t$, and $x_{<t}$ represents the sequence of generated tokens up to the time step $(t-1)$.

**Next Token Decoding.** Based on the predicted probabilities $p(x_t|v, q, x_{<t})$, various decoding strategies—such as beam search, and contrastive decoding (e.g., VCD)—can be flexibly applied to generate output. While these strategies can marginally reduce hallucinations, they are typically restricted to single-model outputs, cannot leverage any external knowledge, and fail to fully exploit complementary strengths across different models. As a result, they remain prone to errors, especially in open-domain scenarios. In contrast, our proposed method adaptively fuses the token-level logits from multiple LVLMs that share the same vocabulary, immediately after the forward pass. By leveraging the diverse capabilities of different models, our approach more effectively mitigates hallucinations in both general-purpose and task-specific settings.

### 3.2 ADAPTIVE TOKEN ENSEMBLE DECODING

To leverage the complementary strengths of diverse LVLMs and enhance general task performance while reducing hallucinations, we propose a token-level ensemble decoding approach. Specifically, we introduce **Adaptive Token Ensemble Decoding (ATED)**, a training-free method that employs an uncertainty-guided weighting fusion strategy to dynamically integrate multiple LVLMs during inference. The overall framework is illustrated in Figure 2.

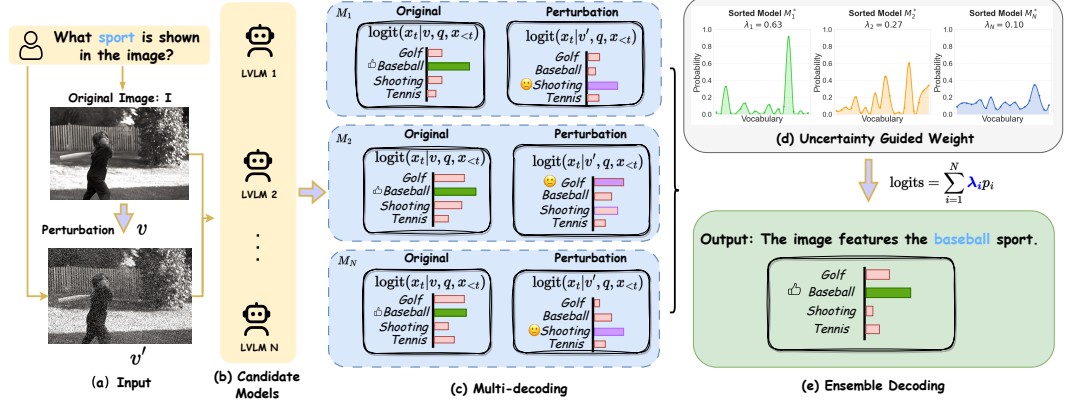

Figure 2: Overall pipeline of **Adapative Token Ensemble Decoding (ATED)**. In module (a), the system takes as input a set of system instructions, the original image, and its perturbed variants, which are passed to multiple candidate LVLMs in module (b). Module (c) generates multiple decoding streams. In module (d), the importance of each LVLM's output is estimated using uncertainty-guided weights and refined via greedy uncertainty optimization. Finally, module (e) aggregates the results through ensemble decoding to produce the final output. This process is dynamically repeated at each time step $t$ during token generation, ensuring both reliability and consistency of the results.

Given a set of LVLMs $\{M_1, \ldots, M_N\}$ at test time, ATED adaptively fuses their output logits by assigning each model an importance weight $\{\lambda_1, \ldots, \lambda_N\}$, where $\lambda_i$ reflects how well model $M_i$ interprets the visual and textual inputs. At each decoding step $t$, each model $M_i$ takes the visual input $v$ and the text query input $q$ along with previously generated tokens $x_{<t}$ to generate a logit score over a shared vocabulary. Assuming all models share the same vocabulary $\boldsymbol{O}$, ATED integrates the logits from all $N$ models to compute the final probabilities as follows:

$$p(x_t|v, q, x_{<t}) = \sum_{i=1}^{N} \underbrace{\lambda_i}_{unknown} p_i, \tag{2}$$

where $i = 1, \ldots, N$ indexes the models, $p_i$ denotes the final decoding logits of model $M_i$. We assume that the weights $\lambda_i \in [0, 1]$ are normalized, i.e., $\sum_{i=1}^{N} \lambda_i = 1$.

Following the approach of Leng et al. (2023), we introduce visual uncertainty by applying Gaussian noise masks to the original image $I$, thereby generating multiple perturbed variants for each input. In the model ensemble process, we further adopt a multi-path contrastive decoding strategy to enhance the model's robustness under visually uncertain conditions. This approach effectively alleviates hallucinations commonly observed in the backbone model when processing ambiguous or degraded visual inputs. Formally, the multi-path decoding can be expressed as:

$$p_i(x_t \mid v, v', q, x_{<t}) = \text{softmax}\Big[(1 + \alpha)\, \text{logit}_\phi\big(x_t \mid v, q, x_{<t}\big) - \alpha\, \text{logit}_\phi\big(x_t \mid v', q, x_{<t}\big)\Big], \tag{3}$$

where $\alpha$ denotes a hyperparameter controlling the intensity of visual contrastive decoding, and $v'$ represents a perturbed version of the original image $I$.

Inspired by Chen et al. (2024a); Qiu et al. (2025); Dey et al. (2025), we propose utilizing the entropy of probability distributions derived directly from visual features as a principled uncertainty metric when LVLMs generate the next textual token. This metric effectively reflects the model's token prediction confidence and the associated adaptive weights $\lambda_i$ under current multimodal inputs. Moreover, the metric is naturally aligned with the training objectives of causal language modeling. Analyzing prediction entropy under different visual conditions enables us to systematically evaluate not only the model's depth of understanding for specific visual content, but also the overall quality of vision-language alignment and the degree of distributional shift between the visual input and the model's training data Gonen et al. (2024). Such cross-modal uncertainty analysis thus provides a a valuable new perspective for rigorously assessing the generalization ability of multimodal large models, particularly under realistic and challenging open-world scenarios.

## 3.3 UNCERTAINTY-GUIDED WEIGHT

**Uncertainty Minimization.** Given the tokenized inputs at decoding step $t$, the uncertainty score $H_i$ for model $M_i$ iis formally defined as the entropy of its output probability distribution over the candidate vocabulary at that time step. The score is computed as follows:

$$H_i = - \sum_{x_t \in \mathcal{O}} p_i \log p_i, \tag{4}$$

where $p_i$ represents the normalized probability corresponding to vocabulary $\boldsymbol{O}$, conditioned on the preceding tokens $x_{<t}$ by model $M_i$.

We formulate the assignment of importance weights across $N$ models as an optimization problem that requires no training or labeled data, and can be solved directly during next token prediction. Formally, our optimization framework is defined as follows:

$$\lambda_1^*, \ldots, \lambda_N^* = \underset{\lambda_1, \ldots, \lambda_N}{\arg\min} - \sum p_i \log p_i, \quad p_i = \text{softmax}(\sum_{i=1}^{N} \lambda_i p_i(x_t|v, v'q, x_{<t})), \tag{5}$$

where the weights $\lambda_i$ are defined to be inversely proportional to each model's normalized uncertainty score—i.e., models exhibiting lower prediction uncertainty are assigned correspondingly higher weights, while those with higher uncertainty contribute less to the final decision. All weights are constrained such that $\sum_{i=1}^{N} \lambda_i = 1$ and $\lambda_i \in [0, 1]$.

**Uncertainty Greedy Optimization.** To address the uncertainty minimization problem proposed in Equation 5, we introduce an efficient greedy optimization algorithm that incrementally ensembles LVLMs. Specifically, we first compute the uncertainty of each LVLM's next-token prediction using Equation 4, and then sort the LVLMs $M_1, \ldots, M_N$ based on their uncertainties scores. Let the sorted models be denoted as:

$$[M_1^*, \ldots, M_N^*] = \text{argsort}(H_1, \ldots, H_N), \tag{6}$$

where $[M_1^*, \ldots, M_N^*]$ are ordered by the lowest to highest uncertainty scores, and set the weight of the top-ranked model to $\lambda_1^* = 1$ and $\lambda_{i>1}^* = 0, i = \{2, \ldots, N\}$.

Subsequently, during the sequential ensemble process, we perform a grid search over the interval $[0, 1]$ with a step size of $s$ for each candidate model to be incorporated, traversing different values of the weight $\lambda$ to identify the optimal value that minimizes the ensemble's uncertainty score. Specifically, the greedy optimization first determines the relative weight between the top-ranked and the second-ranked models. For each candidate value of $\lambda$, we obtain the corresponding probability distribution via softmax, and then compute the uncertainty of the ensemble according to Equation 5. The value of $\lambda_i^*$ that results in the lowest uncertainty is selected as the optimal weight for this round and is used to update and fuse the ensemble output. This greedy optimization procedure is then iteratively applied to all remaining candidate models, dynamically updating the fused logits at each step based on the previous ensemble output.

To avoid redundant computation, we introduce two early stopping criteria: (1) if the optimal weight $\lambda = 1$ in any round, indicating that the current model does not contribute to uncertainty reduction, the subsequent evaluation for that model can be skipped; and (2) if the ensemble's uncertainty score falls below a threshold $\varepsilon$, the iterative process can be immediately terminated to further save computational resources. The algorithm is summarized in Appendix D.

## 3.4 EXPERIMENTAL SETTINGS

### 3.4.1 DATASETS.

We evaluate the performance of our proposed model on three widely used benchmark datasets that encompass diverse multimodal tasks, as detailed below.

**POPE (Probability of Object Presence Estimation).** It is a widely used benchmark dataset introduced by Li et al. (2023d) for evaluating object hallucination in LVLMs, which integrates the MSCOCO Lin et al. (2015), A - OKVQA Schwenk et al. (2022), and GQA Hudson & Manning (2019) datasets to form 27,000 query-answer pairs for evaluation. Performance is quantified using standard metrics, including *accuracy*, *precision*, *recall*, and *F1 score*.

**CHAIR (The Caption Hallucination Assessment with Image Relevance).** It is a dataset from Rohrbach et al. (2019) for evaluating object hallucination in image captioning, which has two main variants: $CHAIR_I$ and $CHAIR_s$, focusing on instance and sentence levels, respectively.

**MME (Multimodal Large Language Model Evaluation).** Fu et al. (2024) proposed the MME, MME, a benchmark evaluating LVLMs on perception and cognition. We assess four sub-tasks: object existence, counting, position, and color. Model performance is measured using the *accuracy+* metric. More evaluation metric details can be found in Appendix C.2.

### 3.4.2 MODELS.

We integrate our proposed method with four popular LVLMs: InstructBLIP Dai et al. (2023), MiniGPT-4 Zhu et al. (2023), LLaVA-1.5 Liu et al. (2024c), and LLaVA-Next Liu et al. (2024d). All the LVLMs used have a language model size of 7 billion parameters (7B). InstructBLIP and MiniGPT-4 utilize a Q-former Li et al. (2023a), which represents an image using only 32 tokens, effectively bridging the visual and textual modalities. LLaVA-1.5 and LLaVA-NeXT employ a linear projection layer to align features from the two modalities. All LVLMs adopt pre-trained vision encoders such as the CLIP vision encoder Radford et al. (2021), along with pre-trained LLMs as language decoders, such as LLaMA Touvron et al. (2023) or Vicuna v1.1 Chiang et al. (2023). Complete experimental details are provided in Appendix C.3.

### 3.4.3 BASELINES.

For the object hallucination evaluation, we employ several widely used decoding strategies, such as multinomial sampling (Default) and four state-of-the-art training-free decoding methods. OPERA Huang et al. (2024) builds upon beam search and alleviates hallucination by penalizing certain patterns of knowledge aggregation. VCD Leng et al. (2023) reduces hallucination by decoding with noisy images in a contrastive manner. ICD Wang et al. (2024) mitigates hallucination by designing negative prompts to interfere with the visual inputs during contrastive decoding. SID Huo et al. (2025) mitigateas hallucinations by introspectively filtering low-relevance visual signals during generation. For all the baselines, we use the default hyperparameters provided by their original source code to ensure a fair comparison. We posit that our method, being LVLM-agnostic, can be easily integrated into various off-the-shelf LVLMs that share the same vocabulary.

## 3.5 EXPERIMENTAL RESULTS

**Results on POPE.** We begin with the most widely adopted benchmark for evaluating object hallucination. Table 1 reports the average performance across three evaluation settings—*random*, *popular*, and *adversarial*—on various datasets, where *Default* refers to the unmodified backbone model. Our evaluation of ATED includes three different configurations: two distinct LVLM ensemble variants ($\mathbf{ATED}^{\&}$ and $\mathbf{ATED}^{\#}$), as well as a version based on the $\mathbf{ATED}^{\#}$ that excludes vision-contrastive decoding ($\mathbf{ATED}^{*}$). For clarity, we highlight the best performances within each setting for each backbone in bold. Compared with each respective backbone, ATED achieves improvements of 4.20%–6.29% in *Accuracy* and 6.29%–6.97% in *F1-score*. Furthermore, on both LLaVA-1.5 and InstructBLIP, ATED consistently surpasses state-of-the-art methods ICD and VCD, attaining additional gains ranging from 0.89% to 5.10% in *Accuracy* and 0.80% to 2.94% in *F1-score*. These results further validate the effectiveness of ATED in mitigating object hallucination.

**Results on CHAIR.** Beyond the binary "yes" or "no" evaluations on the POPE benchmark, we further validate the effectiveness of ATED in open-ended image captioning using the CHAIR metric. Specifically, we randomly sample 500 images from the validation split of the MSCOCO dataset and query various LVLMs with the prompt, "Please describe this image in detail." As shown in Table 2, when setting the max new token length to 64, our proposed ATED method significantly outperforms all baseline decoding approaches on the $CHAIR_S$ metric, achieving improvements of 21.13%-41.24% over the strongest baseline. Notably, when increasing the generation length to 512 tokens, $\mathbf{ATED}^{\#}$ still attains the best performance on the $CHAIR_S$ metric, with an improvement of approximately 30.0%. More results are provided in Appendix E.1.

Table 1: **Comparison of different decoding strategies on POPE.** Results are from the papers or re-implemented based on official codes. Higher values indicate better performance. *Note:* ∗ denotes ATED without vision contrastive decoding, & denotes ensemble with LLaVA-1.5 and InstructBLIP, # denotes ensemble with LLaVA-1.5, InstructBLIP and LLaVA-NeXT.

| Model | Method | Random | | Popular | | Adversarial | |
|---|---|---|---|---|---|---|---|
| | | Accuracy | F1 Score | Accuracy | F1 Score | Accuracy | F1 Score |
| LLaVA-1.5 | Default | 83.86 | 82.68 | 80.82 | 79.54 | 76.42 | 76.61 |
| | OPERA | 88.85 | 88.67 | 82.77 | 83.40 | 79.16 | 80.93 |
| | VCD | 87.20 | 87.17 | 83.08 | 83.07 | 77.70 | 79.14 |
| | ICD | 83.15 | 83.91 | 83.15 | 83.91 | 79.13 | 80.41 |
| | SID | 89.46 | 89.62 | 85.13 | 85.94 | **83.24** | 82.21 |
| InstructBLIP | Default | 81.44 | 81.21 | 79.06 | 79.12 | 76.29 | 76.99 |
| | OPERA | 84.57 | 83.74 | 78.24 | 79.15 | 74.59 | 76.33 |
| | VCD | 84.91 | 84.08 | 81.89 | 81.46 | 79.97 | 79.90 |
| | ICD | 81.12 | 82.25 | 81.12 | 82.25 | 76.82 | 78.99 |
| | SID | 87.23 | 86.90 | 81.16 | 82.57 | 78.51 | 81.26 |
| MiniGPT4 | Default | 65.65 | 66.45 | 59.61 | 62.54 | 58.35 | 62.22 |
| | OPERA | 79.91 | 77.60 | 73.78 | 72.23 | 71.76 | 70.64 |
| | VCD | 67.79 | 68.54 | 62.42 | 65.24 | 60.17 | 63.94 |
| | ICD | 71.89 | 75.63 | 64.58 | 75.33 | 61.77 | 67.61 |
| | SID | 75.20 | 76.12 | 68.94 | 72.93 | 66.57 | 69.40 |
| LLaVA-NeXT | Default | 84.83 | 81.78 | 81.00 | 79.72 | 76.01 | 75.83 |
| | OPERA | 88.41 | 87.33 | 82.69 | 83.48 | 79.22 | 79.40 |
| | VCD | 86.01 | 85.20 | 81.90 | 82.23 | 78.00 | 79.12 |
| | ICD | 82.14 | 82.09 | 81.95 | 81.87 | 79.24 | 78.89 |
| | SID | 89.54 | **89.67** | 85.24 | 85.67 | 82.43 | 81.51 |
| Ensemble | ATED∗ | 88.74 | 87.82 | 83.62 | 84.82 | 78.86 | 81.21 |
| | ATED& | 89.21 | 89.39 | 85.32 | 85.66 | 81.51 | 82.32 |
| | ATED# | **89.83** | 89.35 | **86.71** | **85.97** | 82.96 | **82.78** |

Table 2: **Comparison of different decoding strategies on CHAIR**. Results are from the papers or re-implemented based on official codes, lower values indicate better performance. *Note:* & denotes ensemble with LLaVA-1.5 and InstructBLIP, # denotes ensemble with LLaVA-1.5, InstructBLIP and LLaVA-NeXT.

| Type | Method | LLaVA-1.5 | | InstructBLIP | | MiniGPT4 | | LLaVA-NeXT | |
|---|---|---|---|---|---|---|---|---|---|
| | | $CHAIR_S$ | $CHAIR_I$ | $CHAIR_S$ | $CHAIR_I$ | $CHAIR_S$ | $CHAIR_I$ | $CHAIR_S$ | $CHAIR_I$ |
| Single | Default | 24.8 | 8.9 | 30.3 | 13.9 | 19.8 | 8.5 | 24.3 | 8.5 |
| | OPERA | 21.8 | 8.2 | 28.4 | 9.7 | 22.6 | 8.2 | 21.3 | **7.7** |
| | VCD | 23.6 | 8.6 | 30.0 | 11.2 | 22.0 | 10.6 | 23.3 | 8.3 |
| | ICD | 21.0 | 8.7 | 21.8 | 8.2 | 20.0 | 8.7 | 20.6 | 8.5 |
| | SID | 20.7 | 8.4 | 20.7 | 8.4 | 23.1 | 10.7 | 19.4 | 7.8 |

| Type | Method | ATED& | | ATED# | |
|---|---|---|---|---|---|
| Ensemble | Ours | $CHAIR_S$ | $CHAIR_I$ | $CHAIR_S$ | $CHAIR_I$ |
| | | 15.3 | 10.9 | **11.4** | 8.1 |

**Results on MME.** We extend the evaluation to include hallucinations at the object attribute level. We further conduct a systematic and comprehensive evaluation of the proposed ATED method on the MME hallucination subset, which encompasses both object-level tasks (existence identification and quantity judgment) and attribute-level tasks (location identification and color classification).

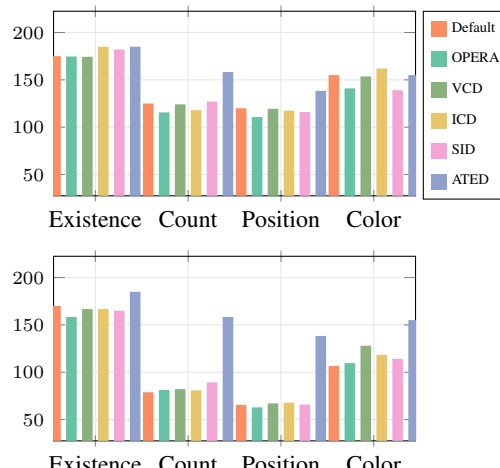 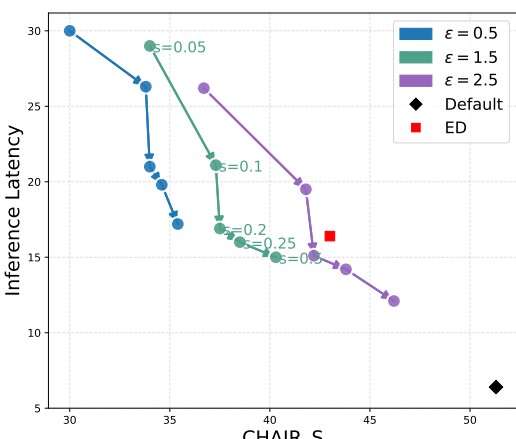

Figure 3: Results on the hallucination subset of MME with decoding strategies: LLaVA-1.5 (top), InstructBLIP (bottom).

Figure 4: Trade-off between inference latency and caption generation performance.

As shown in Figure 3, ATED achieves the highest performance on location questions, and attains highest accuracy on all existence-related questions. Overall, our ATED method significantly outperforms both the default LVLMs and other baseline methods across all four tasks (with *accuracy+* metric improvements of at least +61.7% and +54.2%, respectively).

In addition, we provide several qualitative cases that further demonstrate ATED's strong ability on mitigating hallucinations, extending beyond object hallucination. These cases use the instructions with "Please describe this image in detail.", and more details are provided in Appendix E.6.

## 3.6 ABLATION STUDIES

**Adaptive Uncertainty-Guided Weight.** To further validate the effectiveness of the adaptive uncertainty-guided weighting strategy, we conduct a systematic comparison of multiple weighting schemes on the POPE MSCOCO benchmark. Under the same model combinations and perturbation configurations, we evaluate three approaches: a uniform logit-weighting baseline ("Uniform"), a confidence-based weighting scheme ("Confidence-based"), and uncertainty-guided weighting variants with and without the Uncertainty-Greedy Optimization

Table 3: Results on the POPE MSCOCO benchmark comparing various fusion method.

| Setting | Method | Accuracy | F1 Score |
|---|---|---|---|
| **Random** | Uniform | 87.03 | 86.65 |
| | Confidence-based | 88.13 | 86.76 |
| | ATED (w/o UGO) | **89.30** | 87.78 |
| | ATED | 88.97 | **88.29** |
| **Popular** | Uniform | 84.57 | 84.37 |
| | Confidence-based | 86.27 | 84.79 |
| | ATED (w/o UGO) | 87.10 | 85.98 |
| | ATED | **87.57** | **86.58** |
| **Adversarial** | Uniform | 81.07 | 81.65 |
| | Confidence-based | 84.77 | 83.40 |
| | ATED (w/o UGO) | 85.17 | 84.25 |
| | ATED | **85.37** | **84.64** |

(UGO) module. As shown in Table 3, our proposed ATED method achieves average improvements of 3.66% in *Accuracy* and 2.71% in *F1 Score* over the "Uniform" baseline. For the confidence-based variant with UGO("Confidence-based"), which uses the Maximum Softmax Probability as an alternative uncertainty signal, ATED consistently outperforms this variant across all three settings on the POPE MSCOCO benchmark.

When the UGO module is removed, the performance of the model ensemble degrades to varying degrees, indicating that the absence of uncertainty-aware optimization weakens the effectiveness of the ensemble strategy. These results clearly demonstrate that adaptive uncertainty-guided weighting plays a key role in multimodal model ensembling, and that, when further enhanced by greedy optimization, it can more effectively amplify the performance gains of model ensembles. Overall, our

Table 4: Comparison of performance and inference latency on CHAIR. The best results are highlighted in **bold**, and the second-best results are underlined. ***Note***: # denotes ensemble with LLaVA-1.5, InstructBLIP and LLaVA-NeXT.

| Method | $s$ | $\varepsilon$ | CHAIR_S (512)↓ | Average Length↑ | Inference Latency↓ |
|--------|-----|---------------|----------------|-----------------|---------------------|
| default | – | – | 51.3 | 100.6 | **6.4** |
| ED | – | – | 43.0 | 100.1 | 16.4 |
| ATED# | 0.5 | 2.5 | 40.4 | 103.6 | 12.3 |
| ATED# | 0.2 | 1.5 | **36.2** | **105.2** | 16.5 |

findings provide strong empirical evidence that the proposed adaptive weighting strategy is fundamental for achieving robust and efficient multimodal integration.

In addition, we further investigate the effect of visual perturbations on hallucination mitigation in LVLM ensemble decoding across different tasks to further validate their impact on ensemble performance. More detailed experimental results are provided in Appendix E.5.

### 3.7 INFERENCE LATENCY AND CAPTION GENERATION TRADE-OFF

To evaluate the computational efficiency of ATED, we conduct experiments on the CHAIR benchmark dataset and compare our method against the default model (LLaVA-1.5) and the state-of-the-art ensemble-based baseline ED Cho et al. (2025). The results are summarized in Table 4. In this experiment, we consider three approaches: standard decoding (default), the ensemble decoding baseline ED[1], and several ATED variants. We evaluate them on three metrics: $CHAIR_S$ (with the max new token length set to 512), the average caption length, and the per-image inference latency. The default method does not introduce any additional mechanisms and therefore achieves the shortest inference time (6.4), but its performance on object hallucination tasks is limited. In contrast, ED currently supports only the LLaVA backbone, whose vision branch adopts a Transformer-based architecture such as ViT Dosovitskiy et al. (2020) to partition and encode the input image into patches. While this design can improve performance, it lacks dynamic adaptability and is not easily extensible to multiple models, its inference latency is more than twice that of standard decoding. By comparison, ATED achieves better hallucination-related performance while maintaining inference latency in the same order of magnitude as ED, thus offering a more favorable performance–efficiency trade-off.

We further analyze the trade-off between inference latency and caption generation performance, with detailed results shown in 4. This analysis demonstrates that ATED provides a highly flexible and dynamically adjustable mechanism for balancing latency and accuracy. By appropriately increasing the step size $s$ or the threshold $\varepsilon$, the model can deliberately trade a small amount of accuracy for a substantial reduction in inference latency.

Regarding GPU memory, ATED can be accommodated on a single L20 (48 GB) with all components fully loaded. Therefore, in scenarios where additional training or large GPU clusters are costly or impractical, ATED remains a feasible and relatively lightweight ensemble decoding strategy.

### 3.8 ANALYSIS OF LVLM ENSEMBLE STRATEGIES

To comprehensively evaluate the effectiveness of different LVLM ensemble strategies across various tasks, we conduct ensemble experiments on the POPE and MME benchmarks using LLaVA-1.5, InstructBLIP, and LLaVA-NeXT. The results are summarized in Table 5. Our findings indicate that when the performance gap between models is substantial (e.g., InstructBLIP and LLaVA-1.5 exhibit more than a 10%

Table 5: Comparison of LVLM ensemble strategies performance across the POPE and MME benchmarks.

| Model | POPE | | MME |
|-------|------|------|-----|
| | Accuracy | F1 Score | Accuracy+ |
| InstructBLIP | 78.93 | 79.11 | 1385.87 |
| LLaVA-1.5 | 80.37 | 79.61 | 1715.40 |
| + InstructBLIP (U) | 83.90 | 85.14 | 1437.84 |
| + InstructBLIP | 85.35 | 85.79 | 1718.18 |
| + LLaVA-NeXT | **86.55** | **86.13** | **1788.09** |

difference on the MME benchmark), simple uniform (U) averaging of token probabilities across models not only fails to enhance performance but may even degrade it, as additional noise is introduced by lower-performing models. In contrast, when the gap is relatively small (for instance, LLaVA-1.5's *F1 score* on POPE exceeds that of LLaVA-NeXT by only about 5%), probability averaging can still provide noticeable improvements over individual models. ATED, however, effectively addresses these inherent limitations. In contrast to uniform averaging methods, it leverages an adaptive weighting strategy guided by model-specific uncertainty, thereby ensuring more stable and reliable performance improvements. This design endows ATED with enhanced robustness and broader applicability across a wide range of multimodal tasks.

## 4 CONCLUSION

In this paper, we propose ATED, the first training-free multimodal ensemble decoding method that effectively mitigates hallucinations across diverse multimodal tasks. During inference, ATED performs parallel processing with multiple LVLMs and adaptively fuses token-level logits, enabling finer-grained semantic control and more consistent generation by dynamically adjusting model importance through an uncertainty-guided weighting mechanism. Moreover, ATED offers strong flexibility, allowing users to balance maximum performance and inference efficiency, which makes it well-suited for diverse application scenarios and varying task requirements. Extensive experiments across multiple benchmarks demonstrate that ATED consistently outperforms prior methods, delivering substantial improvements in both accuracy and robustness within vision-language applications.

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

# APPENDIX

## A   THE USE OF LARGE LANGUAGE MODELS(LLMS)

We used a large language model (ChatGPT, GPT-4) solely for English copy-editing (grammar and style). The model was not used to design experiments, analyze data, generate results, figures, or references; all edits were reviewed by the authors, who take full responsibility for the content.

## B   MORE BACKGROUNDS

### B.1   ARCHITECTURE OF VISION-LANGUAGE MODELS

Large Vision-Language Models (LVLMs) integrate pretrained image encoders and large-scale language models to support tasks such as image captioning and visual question answering. Typically, a frozen vision encoder such as CLIP extracts dense image embeddings Radford et al. (2021), which are projected into the language space and fed into a decoder-only LLM like LLaMA. Architectures such as BLIP-2 Li et al. (2023a) and InstructBLIP Dai et al. (2023) adopt this two-tower design and align the modalities using lightweight adapters or learned commands.

**LLaVA-1.5** is a refinement over the original LLaVA model, featuring a simplified architecture and improved training pipeline. It employs CLIP-ViT-L as the vision encoder and a Vicuna-based

decoder-only language model. The two modalities are connected via a trainable MLP projection layer, which maps visual tokens into the language embedding space. Trained with visual instruction tuning on synthetic datasets, it achieves strong results across various benchmarks. Liu et al. (2023)

**InstructBLIP** builds on BLIP-2 by introducing an instruction-aware query transformer that conditions the vision encoder's output on task-specific prompts. It integrates a pretrained Vision Transformer (ViT) with a Q-Former, feeding the encoded visual queries to a language model such as Flan-T5 or Vicuna. It is trained using instruction tuning on a collection of 26 datasets. Dai et al. (2023)

**MiniGPT-4** aims to replicate the capabilities of GPT-4-based vision-language systems using open-source components. It integrates a frozen ViT-based vision encoder with Vicuna, connected via a lightweight linear projection layer. Training involves two stages: pre-alignment on image-text pairs followed by fine-tunning on high quality image descriptions. Its minimal parameter count enables efficient multimodal alignment with strong performance in image captioning. Zhu et al. (2023)

**LLaVA-NEXT** is an enhanced version of LLaVA-1.5, optimized for higher visual reasoning fidelity. It retains the MLP projection structure but augments training with improved instruction-following datasets and higher-resolution visual inputs. It achieves better performance in OCR, compositional reasoning, and world knowledge benchmakrs. Liu et al. (2024d)

## B.2 ENSEMBLE LEARNING IN NLP

Ensemble learning has long been a reliable strategy in machine learning to improve robustness, reduce overfitting, and enhance generalization. By combining the predicitions of multiple models or decision rules, ensembles can correct individual biases and reduce the variance of outputs Dietterich (2000), Classical ensemble methods include bagging, boosting, and stacking, all of which have demonstrated strong performance in classification tasks such as sentiment analysis, topic classification, and named entity recognition Opitz & Maclin (1999).

In the domain of Natural Language Processing (NLP), ensemble methods have been applied extensively in both structured prediction and generation tasks. For example, ensemble decoding, which involves averaging or voting across multiple language models. has been shown to improve fluency and factuality in neural machine translation Sennrich et al. (2016). Recent work has also explored ensemble inference for large language models, aggregating outputs per-token-level logits from multiple sources to improve consistency and reduce hallucination

In multimodal learning, ensemble approaches are gaining traction as a decoding-level intervention. Rather than relying on a single model's output, ensembles constructed across different model variants or decoding stratigies can better capture complementary evidence, making them suitable for suppressing the hallucination in vision-language tasks.

## C EXPERIMENTAL SETTINGS

### C.1 DETAILS ABOUT BASELINE

To mitigate hallucination without retraining, a variety of decoding-time techniques have been proposed:

- **ICD (nstruction-Contrastice Decod- ing)** Wang et al. (2024). ICD leverages instruction-level perturbations to reduce hallucination in multimodal large language models. It operates by introducing minimal semantic alterations to the input prompt, such as inserting irrelevant phrases or modifiying the question structure, and then comparing the model's output distributions under both the original and perturbed instructions. Tokens exhibiting instability across these variants are identified as potentially hallucinated and are downweighted during generation.

- **VCD (Visual-Contrastive Decoding)** Leng et al. (2023). VCD aims to improve visual consistency by introducing small-scale perturbations to the visual input and contrasting the model's responses. The approach applies controlled distortions, such as Gaussian blur, occlusion, or token masking, to the image embeddings and measures output divergence. To-

kens highly sensitive to such perturbations are treated as visually fragile and are penalized during decoding.

- **OPERA (Overtrust Penalty with Retrospective Adjustment)** Huang et al. (2024). OPERA introduces a two-stage mechanism to address hallucination in multimodal generation: overtrust penalty and retrospective adjustment. During decoding, it applies a regularization term to suppress overconfident token predictions that exhibit weak visual grounding. After generation, a retrospective evaluation is performed to re-rank or adjust outputs based on their semantic agreement with the image.

- **SID (Self-Introspective Decoding)** Huo et al. (2025). SID mitigateas hallucinations by introspectively filtering low-relevance visual signals during generation. It evaluates the contextual alignment of visual tokens with both the preceding textual context and the decoding history, retaining only those with strong semantic relevance. By pruning distractive or semantically weak visual features early in decoding, SID improves grounding accuracy, particularly in complex or visually dense scenarios.

- **Ensemble Decoding (ED)** Cho et al. (2025). ED combines multiple generation pathways to improve robustness and reduce hallucination. It operates by aggregating outputs from a set of models or decoding configurations, such as different random seed, visual crops, or temperature settings, and fusing them through majority voting, logit averaging, or response re-ranking. This ensemble process helps to mitigate the influence of unstable or outlier predictions by emphasizing consensus across multiple decoders.

- **RLAIF-V** Yu et al. (2025). RLAIF-V aligns multimodal large models within a fully open-source paradigm. It constructs high-quality feedback data for preference learning and employs self-feedback during inference to enable scalable reasoning, thereby yielding a vision-language model whose preference alignment is driven by large-scale open-source AI feedback.

All these methods operate without modifying model parameters, offering flexible, training-free solutions for enhancing visual faithfulness during inference.

## C.2 EVALUATION METRIC DETAILS

**The Polling-based Object Probing Evaluation (POPE)** benchmark is a systematic framework designed to assess object hallucination in Large Vision-Language Models (LVLMs) during image description tasks. POPE employs a binary question-answering format, using prompts such as "Does the image contain ___?" to evaluate a model's ability to accurately determine the presence or absence of specific objects within images. To construct negative samples—instances where the object is absent from the image—POPE utilizes three distinct strategies: random sampling involves selecting objects that do not appear in the image at random; popular sampling selects absent objects from a pool of frequently occurring objects across the dataset; adversarial sampling prioritizes objects that commonly co-occur with present objects but are absent in the current image. The benchmark integrates three datasets: MSCOCO, A-OKVQA, and GQA. From each dataset, 500 images are selected, and six questions are generated per image, resulting in a total of 27,000 query-answer pairs for evaluation. Performance is measured using standard metrics, including accuracy, precision, recall, and F1 score, with higher values indicating a model's superior capability in mitigating hallucinations such as fabricated objects and erroneous descriptions.

**The Caption Hallucination Assessment with Image Relevance (CHAIR)** metric is a specialized evaluation framework designed to quantify object hallucination in image captioning models. CHAIR assesses the alignment between generated captions and the actual visual content by comparing the objects mentioned in the captions against ground-truth annotations from datasets like MSCOCO.

$$C_S = \frac{|\{\text{hallucinated objects}\}|}{|\{\text{all mentioned objects}\}|} \tag{7}$$

$$C_I = \frac{|\{\text{captions w/ hallucinated objects}\}|}{|\{\text{all captions}\}|} \tag{8}$$

The metric comprises two variants: CHAIRi (instance-level) and CHAIRs (sentence-level). CHAIRi calculates the proportion of hallucinated object mentions relative to all object mentions in the generated captions, while CHAIRs measures the fraction of sentences that contain at least one hallucinated

object. Lower values in both metrics indicate better performance in mitigating object hallucinations .

**The Multimodal Model Evaluation (MME)** benchmark offers a comprehensive framework for assessing Large Vision-Language Models (LVLMs) across a spectrum of tasks, encompassing both perceptual and cognitive dimensions. Specifically, MME comprises ten perception-oriented subtasks and four cognition-focused ones, facilitating a holistic evaluation of LVLM capabilities .In the context of object-level hallucination evaluation, MME includes dedicated subsets targeting the "existence" and "count" tasks. The "existence" task assesses a model's ability to accurately identify the presence or absence of specific objects within an image, while the "count" task evaluates the model's proficiency in determining the correct number of instances of a given object.These tasks is quantified using a combined metric of accuracy and accuracy+. Accuracy measures the proportion of correct predictions, while accuracy+ accounts for near-correct responses.

### C.3    IMPLEMENTATION DETAILS

In all experimental settings, the hyper-parameter $\alpha$ is fixed at 1. For visual perturbations in the model ensemble, we adopt a noise-injection strategy, setting the noise steps $T$ to 200 for MME, 500 for LLaVA-Bench, and 999 for POPE. For OPERA, VCD, and SID, we use the default settings as specified in their original papers. We set $s = 0.05$ in uncertainty greedy optimization.

## D    UNCERTAINTY GREEDY OPTIMIZATION

We present the Uncertainty Greedy Optimization algorithm for ATED in Algorithm 1.

---

**Algorithm 1** Uncertainty-Guided Greedy Optimization.

---

**Input:** $P = (v, v', q)$, candidate LVLMs $\{M_1, \ldots, M_N\}$.
**Output:** Uncertainty-Guided Weights $\{\lambda_1^*, \ldots, \lambda_N^*\}$ for input $P$ and candidate LVLMs.
1: **for** $i = 1$ to $N$ **do**
2:      Compute uncertainty score $H_i$.
3: **end for**
4: Sort LVLMs via:
        $[M_1^*, \ldots, M_N^*] \leftarrow \mathrm{argsort}(H_1, \ldots, H_N)$.
5: Initialise weights: $\lambda_1^* \leftarrow 1, \ \lambda_{j>1}^* \leftarrow 0$.
6: **for** $i = 1$ to $N$ **do**
7:      **for** $\lambda \in [0, 1, s = 0.05]$ **do**          *greed search*
8:           Compute logits $p_i$ using $M_i^*$, where

9:           $$p_i = \mathrm{softmax}(\sum_{i=1}^{N} \lambda_i \, p_i(x_t|v, v', q, x_{<t})),$$

10:     **end for**
11:     Set $\lambda_1^*, \ldots, \lambda_N^* = \arg\min_{\lambda_1, \ldots, \lambda_N} -\sum p_i \log p_i$,
12:     Start with $\lambda_1^* = 1$, compute $p_i \leftarrow \lambda p_i + (1-\lambda)p_{i+1}, \lambda^* \leftarrow \lambda \cdot \lambda_{<i}^*$. *weight update and fusion*
13: **end for**
14: **Generation:** ATED integrates the logits from all N LVLMs: $\mathrm{logits} = \sum_{i=1}^{N} \lambda_i p_i$.

---

## E    MORE DETAILED COMPARISON

### E.1    MORE RESULTS ON POPE AND CHAIR

The hyperparameter *max new tokens*, which controls the maximum length of generated responses, plays a critical role in CHAIR-based evaluation. In the main text, we report results using a setting of *max new tokens* = 64. Additional results under a relaxed constraint of *max new tokens* = 512 are provided in Table 6. As Table illustrates, the generation length limit has a substantial impact on LVLM performance under the CHAIR metric. when the token budget is increased from 64 to 512, our method consistently outperforms all baselines on the metric CHAIR$_S$, highlighting its robustness

Table 6: **CHAIR evaluation results on different decoding strategies**. Results are from the papers or re-implemented based on official codes. lower values indicate better performance. ***Note:*** & denotes ensemble with InstructBLIP, # denotes ensemble with InstructBLIP and LLaVA-NeXT.

| Type | METHOD | LLaVA-1.5 | | InstructBLIP | | MiniGPT4 | | LLaVA-NeXT | |
|---|---|---|---|---|---|---|---|---|---|
| | | $CHAIR_S$ | $CHAIR_I$ | $CHAIR_S$ | $CHAIR_I$ | $CHAIR_S$ | $CHAIR_I$ | $CHAIR_S$ | $CHAIR_I$ |
| Single | Default | 51.3 | 16.8 | 55.6 | 24.2 | 33.6 | 19.4 | 42.6 | 14.1 |
| | OPERA | 46.4 | 13.0 | 47.1 | 12.4 | 26.4 | 10.7 | 39.4 | 11.8 |
| | VCD | 51.7 | 15.6 | 51.0 | 16.7 | 30.4 | 14.2 | 41.1 | 12.9 |
| | ICD | 47.4 | 13.9 | 46.3 | 15.3 | 32.6 | 13.1 | 42.1 | 12.6 |
| | SID | 44.2 | 12.2 | 42.3 | 12.4 | 28.5 | **11.7** | 40.8 | 13.0 |
| | ED | 43.0 | 14.0 | - | - | - | - | - | - |
| Ensemble | **Ours** | $ATED^{\&}$ | | | | $ATED^{\#}$ | | | |
| | | $CHAIR_S$ | | $CHAIR_I$ | | $CHAIR_S$ | | $CHAIR_I$ | |
| | | - | | - | | **34.0** | | 17.1 | |

Table 7: Comparison of total accuracy+ cacross different methods on LLaVA-1.5.

| Method | Accuracy+ |
|---|---|
| Default | 1715.40 |
| OPERA | 1773.52 |
| VCD | 1756.02 |
| ICD | 1749.43 |
| SID | 1770.43 |
| **$ATED^{\#}$** | **1788.09** |

Table 8: Evaluation results on POPE and MME with varying noise levels.

| Noise | POPE | | MME |
|---|---|---|---|
| | Accuracy | F1 Score | Accuracy+ |
| **ATED(0)** | 85.90 | 84.19 | 616.67 |
| **ATED(200)** | 87.04 | 85.86 | **636.67** |
| **ATED(500)** | 86.73 | 85.44 | 608.33 |
| **ATED(700)** | 86.88 | 85.49 | 591.67 |
| **ATED(999)** | **87.17** | **86.59** | 576.67 |

and adaptability under varying generation lengths. Furthermore, our model produces responses with an average length of 107.4 tokens as shown in Table 14 , indicating that the observed reduction in object hallucinations is achieved without compromising the richness of the generated descriptions.

We further conduct a systematic comparison between ATED and ED on the POPE benchmark under the same experimental setup as the original ED paper. Overall, ATED achieves a higher average Accuracy than ED (**86.50** vs. **86.31**), and also yields a better average F1 score (**86.03** vs. **85.86**).

In addition, we provide a quantitative analysis of weight allocation during the ensemble process on the POPE benchmark. For this benchmark, we first identify the strongest single-model LVLM (for example, LLaVA-NeXT on POPE), then run ATED& and ATED# on the full evaluation set and record the weight distribution at each decoding step. The results show that approximately 75% of decoding steps assign the highest weight to this strongest model. This indicates that our uncertainty-guided mechanism tends to favor the better-performing model overall, while still dynamically incorporating useful information from other models when they are more confident about specific tokens.

Table 9: Comparison with RLAIF-V and other decoding strategies on the POPE, MME, and MM-Vet benchmarks.

| Method | POPE (R) | | POPE (P) | | POPE (A) | | MME | MMVet |
|---|---|---|---|---|---|---|---|---|
| | Accuracy↑ | F1 Score↑ | Accuracy↑ | F1 Score↑ | Accuracy↑ | F1 Score↑ | Total Score↑ | Overall↑ |
| Default | 83.86 | 82.68 | 80.82 | 79.54 | 76.42 | 76.61 | 1715.40 | 21.6 |
| OPERA | 88.85 | 88.67 | 82.77 | 83.40 | 79.16 | 80.93 | 1773.52 | 26.4 |
| VCD | 87.20 | 87.17 | 83.08 | 83.07 | 77.70 | 79.14 | 1756.02 | 25.5 |
| ICD | 83.15 | 83.91 | 83.15 | 83.91 | 79.13 | 80.41 | 1749.43 | 26.0 |
| SID | 89.46 | 89.62 | 85.13 | 85.94 | 83.24 | 82.21 | 1770.43 | 26.1 |
| **RLAIF-V** | 88.58 | 87.73 | 84.17 | 83.94 | 81.73 | 82.01 | 1768.42 | 25.3 |
| **ATED&** | 89.21 | 89.39 | 85.32 | 85.66 | 81.51 | 82.32 | 1780.69 | 25.0 |
| **ATED#** | 89.83 | 89.35 | 86.71 | 85.97 | 82.96 | 82.78 | 1788.09 | 26.6 |

Table 10: Comparison with different decoding strategies on MM-Vet.

| Model | Method | Overall↑ |
|---|---|---|
| | default | 25.2 |
| | VCD | 25.4 |
| LLaVA-1.5 | OPERA | 25.5 |
| | SID | 26.0 |
| | RLAIF-V | 26.1 |
| **ATED# (w/o UGO)** | – | 25.9 |
| **ATED#** | – | **26.6** |

## E.2 MORE RESULTS ON MME

ATED is designed to integrate the expertise of multiple models, thereby bridging the hallucination gap that exists among different LVLMs during inference. To further investigate whether our approach not only preserves but also potentially enhances the fundamental perception and reasoning capabilities of LVLMs across a broader range of multimodal tasks, we also analyze the comprehensive performance on the MME benchmark, which consists of 14 sub-tasks for evaluating perception and recognition. As shown in Table 7, our method ($Ours_{\#}$) significantly outperforms all baseline approaches based on the LLaVA-1.5 backbone, surpassing both the original LVLMs and the best-performing baselines by a substantial margin (+18.34). These results indicate that our approach not only effectively manages hallucination during inference but also improves the accuracy of the underlying LVLMs on fundamental tasks.

We further conduct a quantitative analysis on the MME benchmark. Specifically, we first identify LLaVA-1.5 as the strongest single-model LVLM on MME, then run ATED and ATED on the full evaluation set and record the weight allocation at each decoding step. The results show that roughly 82% of decoding steps assign the highest weight to this strongest model, which closely mirrors the uncertainty-guided weighting pattern observed on POPE.

## E.3 COMPARISONS WITH RLAIF-V

We use the LLaVA-1.5-7B–based RLAIF-V model for evaluation. Specifically, we evaluate this RLAIF-V model on the POPE(R), MME(P), and MM-Vet Yu et al. (2023) benchmarks, and report the results in Table 9 alongside our ATED variants and other training-free decoding baselines. The results show that, on hallucination-related metrics across all three benchmarks, our ATED variants achieve performance that is comparable to or even better than RLAIF-V. At the same time, ATED remains a training-free decoding strategy that does not require any additional fine-tuning or preference optimization, and thus introduces no extra training cost. Therefore, in scenarios where additional training is expensive or infeasible, ATED can serve as a lightweight and practical alternative.

Table 11: Comparison of different decoding strategies on TextVQA.

| Model | Method | TextVQA val (Acc.↑) |
|---|---|---|
| LLaVA-1.5 | default | 45.5 |
| | VCD | 47.3 |
| | OPERA | 46.6 |
| | RLAIF-V | 43.1 |
| Ensemble | **ATED# (w/o UGO)** | 46.7 |
| Ensemble | **ATED#** | **47.5** |

Table 12: MMMU evaluation results on different decoding strategies.

| Model | Method | Validation Overall↑ |
|---|---|---|
| LLaVA-1.5 | Default | 33.3 |
| | VCD | 34.1 |
| | SID | 34.4 |
| InstructBLIP | Default | 32.9 |
| | VCD | 33.4 |
| | SID | 34.1 |
| **ATED*** | – | 33.9 |
| **ATED&** | – | 34.3 |
| **ATED#** | – | **35.9** |

### E.4 MORE MULTIMODAL EVALUATION

we conduct supplementary experiments on MM-Vet, TextVQA, and MMMU to systematically compare multiple decoding strategies. As shown in Table 10, ATED outperforms RLAIF-V on MM-Vet. For the TextVQA validation set, we unify all methods under the same backbone, LLaVA-1.5-7B based on Vicuna-v1.5-7B, to ensure a fair comparison; the results (Table 11) show that ATED consistently achieves higher accuracy than the default decoding, whereas the preference-aligned model RLAIF-V suffers a performance drop of about 5.3%. On the MMMU validation set (900 images), the results (Table 12) indicate that ATED, ATED&, and ATED# perform on par with or better than the strong SID baseline. In particular, ATED# attains the highest score of 35.9, clearly surpassing LLaVA-1.5 and InstructBLIP under the default, VCD, and SID settings, while ATED& and ATED also outperform both baselines under their default settings and, in some cases, match or exceed SID. Overall, ATED effectively mitigates hallucinations without sacrificing multimodal reasoning ability, and achieves stronger overall performance on multiple benchmarks.

### E.5 IMPACT OF VISION PERTURBATIONS

We further investigate the impact of visual perturbations on hallucination reduction in LVLM ensemble decoding across different tasks. Specifically, we conduct systematic experiments on the POPE-MSCOCO and MME benchmarks to evaluate the performance of dynamic model ensembles under various conditions, including the absence of visual perturbations (**ATED(0)**) and different levels of perturbation intensity. Experimental results in Table 8 demonstrate that, without adaptation to visual perturbations, the performance of multimodal ensemble reasoning significantly degrades on both the POPE and MME datasets—for example, *Accuracy* decreases by 1.4%, F1-*score* drops by 2.7%, and *Accuracy+* decreases by 20. These findings further highlight that introducing multi-path contrastive decoding under visual perturbations can effectively mitigate hallucinations and enhance reasoning performance.

Table 13 presents the quantitative evaluation results of the model under different $\alpha$ values on object-level metrics (Existence, Count), attribute-level metrics (Position, Color), and the overall accuracy (Total Accuracy+). As $\alpha$ increases from 0.5 to 1.0, all metrics demonstrate varying degrees of improvement, with Color showing the most substantial gain—from 140 to 155. These improvements are reflected in the Total Accuracy+, which rises from 595.00 to 636.67 as $\alpha$ increases. Moreover, we observe that attribute-level metrics are more sensitive to changes in the intensity of vision-contrastive

regularization compared to object-level metrics, resulting in greater improvements. This finding indicates that appropriately tuning the $\alpha$ parameter not only enhances the model's ability to confirm object information during adaptive ensemble inference but also significantly improves its capability to capture fine-grained attribute details. As a result, the overall prediction accuracy and robustness are further strengthened.

Table 13: Quantitative results on Object-level (Existence, Count), Attribute-level (Position, Color), and Total Accuracy+ for using various noise steps.

| $\alpha$ | Object-Level | | Attribute-Level | | Total Accuracy+ |
|---|---|---|---|---|---|
| | Existence | Count | Position | Color | |
| 0.5 | 180.00 | 143.33 | 131.67 | 140 | 595.00 |
| 0.7 | 180.00 | 143.33 | 136.67 | 140 | 600.00 |
| 1.0 | 185.00 | 158.33 | 138.33 | 155 | 636.67 |

Table 14: Comparison of CHAIR performance across different methods in terms of output length on LLaVA-1.5.

| Method | Length |
|---|---|
| Default | 100.6 |
| OPERA | 98.6 |
| VCD | 100.4 |
| ICD | 106.3 |
| **ATED**$^{\#}$ | **107.4** |

### E.6 Qualitative Analysis

To further evaluate whether ATED effectively mitigates hallucinations beyond quantitative metrics in open-ended generation tasks, we conducted a qualitative analysis on the MSCOCO dataset, using several decoding strategies as baselines. The LVLMs are prompt with "Please describe this image in detail", with the maximum token limit set to 150. As illustrated in Figure 9 and Figure 10, baseline methods including the default decoding, OPERA, and VCD often produce hallucinated content (highlighted in red). In contrast, ATED dynamically selects and weights token-level outputs from multiple models at each decoding step, guided by a greedy uncertainty-minimization strategy. This enables the model to better adapt to contextual environments and significantly improves the credibility and robustness of the generated content.

In addition, we perform GPT-assisted evaluation on the LLaVA-Bench benchmark (Liu et al., 2023). Following evaluation protocol proposed by Yin et al. (2024); An et al. (2025), the model is presented with an image and two candidate descriptions, structured according to the prompt format shown in Figure 6. The GPT-4o API is employed to evaluate the generated responses in terms of factual accuracy (Accuracy) and descriptive richness (Detailedness).

Furthermore, we conducted an additional evaluation based on GPT-4, following the methodology outlined in (Zhao et al., 2023). Specifically, we randomly sampled 200 images from the Visual Genome (VG-100K) dataset (Krishna et al., 2017) and assessed model performance by comparing the generated descriptions with the region descriptions associated with each image. This comparison allows for effective identification of hallucinated content based on semantic inconsistencies. We comprehensively analyzed five key metrics: sentences per image (SPI), words per image (WPI), hallucinated sentence ratio (HSR), hallucinated word ratio (HWR), and mean hallucination ratio (MHR). Notably, higher SPI and WPI values, as well as lower HSR, HWR, and MHR, indicate better model performance. In the radar charts, a larger area reflects superior performance. Multiple models and decoding strategies were included as baselines for comparison. The detailed results are presented in Figure 5. As shown, the proposed ATED method substantially reduces hallucination and effectively suppresses misleading content during generation.

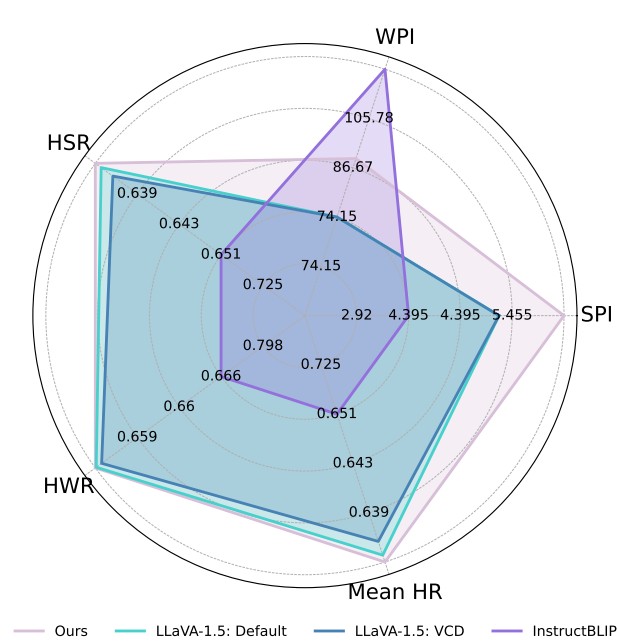

Figure 5: GPT-4 assisted hallucination evaluation.

---

**GPT-4o Prompt**

```
You are an AI designed to evaluate and score the performance of two AI
assistants in describing a given image. Your primary focus is on the
accuracy and detailedness of their descriptions. You will assess the
accuracy by checking for hallucinations—any part of the description
that is inconsistent with the image content. For detailedness, you
will consider how rich the response is in necessary details, excluding
any hallucinated parts. You will provide scores on a scale from 1 to
10 for each assistant separately, based on these criteria. After
scoring, you will offer an explanation for your evaluation, ensuring
it is free from bias and not influenced by the order of presentation
of the responses.

Input format:

[Assistant 1]

{}

[End of Assistant 1]

[Assistant 2]

{}

[End of Assistant 2]

Output format:

Accuracy:

Scores of the two answers:

Detailedness:

Scores of the two answers:
```

Figure 6: GPT-4o prompt format.

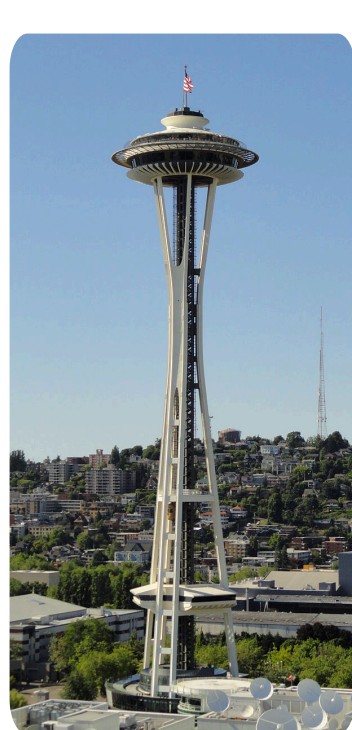

GPT-4o Evaluation: LLaVA-1.5
(OPERA)
Accuracy: 4/10
Detailedness: 5/10

GPT-4o Evaluation: Ours
Accuracy: 9/10
Detailedness: 7/10

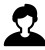

Instr: Please describe this image
in detail

Max New
Token = 150

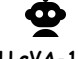

LLaVA-1.5
OPERA

The image depicts a large, white skyscraper towering over a city with a clear blue sky in the background. The skyscraper is located in the center of the city, surrounded by other buildings of various heights. The city is bustling with activity, as evidenced by the presence of numerous cars and trucks on the streets below the skyscraper.

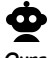

Ours

The image you've shared is a close-up of a tall, slender tower. The tower is predominantly white, with a few black accents that add a touch of contrast. The tower is situated in a city, as evidenced by the buildings and structures that surround it.

*Accuracy:* Assistant 1: This description contains significant hallucinations. The structure in the image is the Space Needle in Seattle, which is not a "large, white skyscraper" and does not resemble one. It is a single, iconic observation tower with a distinct design. The claim of "numerous cars and trucks on the streets" is also inaccurate, as there are no visible streets bustling with traffic in the image. Assistant 2: This response is mostly accurate. The description identifies the subject as a "tall, slender tower" that is "predominantly white with a few black accents," which matches the appearance of the Space Needle. It also matches the setting by mentioning the city with surrounding buildings.

*Detailedness:* Assistant 1: While this response attempts to provide a more elaborate description of the setting, most of the details are fabricated and therefore detract from the evaluation. It does not mention defining features of the tower, such as its observation deck, tapered legs, or unique aesthetic. Assistant 2: This description is more concise but relatively to the point. The mention of black accents and the general shape of the tower improve its detailedness. However, it omits several crucial features, such as the observation deck, the flag, and the surrounding greenery in view.

Figure 7: Qualitative cases on LLaVA-Bench. The hallucinated content is highlighted in red.

Instr: Please describe this image in detail

Max New Token = 150

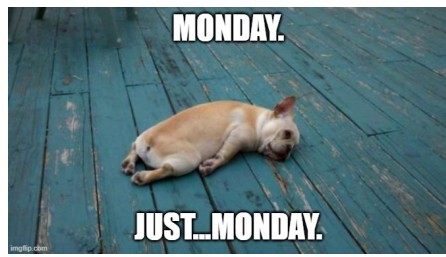

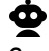 LLaVA-1.5 VCD

A cute little dog is laying down on a wooden floor, resting its head on its paws. The dog is white and brown in color and appears to be sleeping on a blue-colored surface.

Ours

The image features a small dog with a predominantly white coat, lying on its side on a wooden surface. The dog appears to be resting or sleeping, with its head comfortably resting on the wooden surface. The background is out of focus, but it seems to be an indoor setting.

GPT-4o Evaluation: LLaVA-1.5 (VCD)
Accuracy: 4/10
Detailedness: 6/10

GPT-4o Evaluation: Ours
Accuracy: 9/10
Detailedness: 7/10

*Accuracy:* Assistant 1 incorrectly described the dog as "white and brown" and stated that it was "sleeping on a blue-colored surface" which is inaccurate. The dog's coat is light tan with slight white markings, and it is lying directly on a blue wooden floor, not "a blue-colored surface" like a blanket. Additionally, Assistant 1 claimed the dog was "resting its head on its paws" which is not the case—its head is resting on the floor. Assistant 2, on the other hand, provided an accurate description of the dog and its environment, correctly identifying that the dog is lying on its side on the blue wooden floor and avoiding any inaccurate claims.

*Detailedness:* Assistant 1 included some descriptive details, such as identifying the dog's coloration (albeit inaccurately) and the mention of the "blue-colored surface". However, these inclusions were not meaningful due to inaccuracies. Assistant 2 provided sufficient and accurate details without adding unnecessary or hallucinated elements, such as describing the dog, its position, and the surrounding background accurately. Assistant 2's response is slightly more detailed due to a richer contextual hint about the background being "out of focus" and the scene being "an indoor setting."

Figure 8: Qualitative cases on LLaVA-Bench. The hallucinated content is highlighted in red.

**LLaVA-1.5 (VCD):** The image features a woman lying on a bed, with her head resting on her hand. She appears to be looking at the camera, possibly posing for a picture. The bed is covered with a striped blanket, adding a cozy touch to the scene.

**LLaVA-1.5 (OPERA):** The image features a beautiful young woman sitting on a couch, wearing a white shirt. She appears to be posing for the camera, possibly for a magazine or advertisement. The couch she is sitting on has a striped pillow, adding a touch of color to the scene.

**InstructBLIP (OPERA):** The image features a young woman lying on a bed, wearing a white shirt and a red lipstick. She is positioned in the middle of the bed, with her hair cascading down her back. There are two pillows visible in the scene, one on the left side of the bed and

**Ours:** The image features a beautiful young woman with long, dark hair. She is lying down on a bed, dressed in a white shirt. The bed she is on is covered with a white comforter.

**Instr: Please describe this image in details**

Max new token = 64

Figure 9: Qualitative cases on MSCOCO. The hallucinated content is highlighted in red.

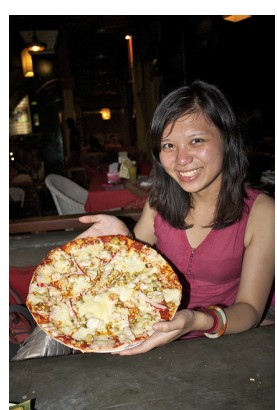

**Instr: Please describe this image in details**

`Max new token = 64`

*LLaVA-1.5 (VCD):* The image features a woman sitting at a dining table, holding a large pizza in her hands. She is smiling and appears to be enjoying the moment. The pizza is placed in the center of the table, and the woman is positioned on the left side of the table."

*LLaVA-1.5 (OPERA):* The image features a woman sitting at a dining table, holding up a large pizza with a smile on her face. The pizza is placed in front of her, occupying a significant portion of the table. There are several other people in the scene, some sitting and others standing around the dining

*InstructBLIP (OPERA):* The image features a woman sitting at a dining table, holding a large pizza in her hands. She is smiling and appears to be eager to share the pizza with others. There are several chairs placed around the table, suggesting that there are other people present.

*Ours:* The image features a woman sitting in front of a dining table. She is holding a large pizza in her hands, which is placed on the table. The woman appears to be in the process of serving the pizza, as she is holding it up.

Figure 10: Qualitative cases on MSCOCO. The hallucinated content is highlighted in red.

