# OpenReview forum: "Beyond Single Models: Mitigating Multimodal Hallucinations via Adaptive Token Ensemble Decoding"
_ICLR.cc/2026/Conference — Submitted to ICLR 2026_

### Official Review · Reviewer_oHTM · 2025-10-28

**Soundness:** 3
**Presentation:** 3
**Contribution:** 3
**Rating:** 6
**Confidence:** 3

**Summary:**

This paper proposes Adaptive Token Ensemble Decoding (ATED), a training-free framework for mitigating object hallucination in Large Vision-Language Models (LVLMs).
ATED aggregates token-level logits from multiple LVLMs via an uncertainty-guided weighting mechanism that dynamically adjusts each model’s contribution during decoding.
A greedy uncertainty-minimization algorithm (UGO) further refines ensemble weights to balance reliability and computational cost.
Across POPE, CHAIR, and MME benchmarks, ATED consistently outperforms strong baselines, reducing hallucinations without harming fluency.
Ablation studies confirm that adaptive weighting and visual perturbations are key to robustness, while latency experiments show favorable trade-offs between inference speed and caption quality.
Overall, ATED demonstrates a flexible and effective approach to improving LVLM robustness in multimodal reasoning and captioning tasks.

**Strengths:**

- The paper addresses an important and persistent weakness of LVLMs (object hallucination) with a conceptually simple yet broadly applicable ensemble framework.
- ATED can be integrated with various off-the-shelf LVLMs (InstructBLIP, MiniGPT-4, LLaVA-1.5/Next) without retraining, showing practical scalability.
- The experiments span multiple benchmarks (POPE, CHAIR, MME) and include both quantitative and qualitative analyses, providing convincing empirical support.

**Weaknesses:**

- The proposed ATED appears conceptually close to prior methods: ensemble decoding (ED) and visual-contrastive decoding (VCD).
The only substantial novelty lies in the Uncertainty-Greedy Optimization (UGO), whose ablation (Table 3) suggests only marginal contribution, making the methodological innovation relatively weak.

- The paper lacks statistical significance analysis or variance reporting, which limits confidence in the reported percentage improvements.

- While ATED claims to be “training-free,” the computation overhead of multi-model forward passes is non-trivial; the paper does not provide detailed latency or cost comparisons under identical hardware constraints.

**Questions:**

- How many LVLMs were actually ensembled in each experiment, and how does performance scale with the number of models (N = 2 → 3 → 4)?

- How sensitive is ATED to the hyperparameters?

- Can the uncertainty-guided weighting be estimated within a single model using internal heads or adapters, rather than across multiple LVLMs?

---

> ### Author Response · Authors · 2025-11-22
> **Response to Reviewer oHTM (part 1/3)**
>
> We greatly appreciate the reviewer’s valuable and insightful feedback. Below, we have provided detailed responses to each of your questions.
>
> > W1. The proposed ATED appears conceptually close to prior methods: ensemble decoding (ED) and visual-contrastive decoding (VCD). The only substantial novelty lies in the Uncertainty-Greedy Optimization (UGO), whose ablation (Table 3) suggests only marginal contribution, making the methodological innovation relatively weak.
>
> **Answer:** The major differences between the proposed ATED and existing approaches, like ED and VCD are as follow: ATED unifies multi-model and multi-path logits within a token-level, uncertainty-driven fusion framework, while ED employs fixed aggregation on a single backbone (e.g. LLaVA) and VCD employs a single-model with two-path contrastive decoding. In addition, the ablation results on POPE MSCOCO **(see Table 3 in the paper)** show that **ATED yields consistent F1 score improvements** across the Random / Popular / Adversarial subsets, with the gains being especially pronounced on the more challenging Popular and Adversarial splits.
>
> &nbsp;
>
> > W2. The paper lacks statistical significance analysis or variance reporting, which limits confidence in the reported percentage improvements.
> >
> **Answer:** To address the concern, we conducted multi-run experiments across hallucination benchmarks (POPE  and CHAIR). For each dataset, we repeated the experiments five times with different random seeds and computed the variance and standard deviation of all key evaluation metrics, as shown in Table 1 below. All results are averaged over five independent runs with different random seeds, and the corresponding p-values are reported for the key comparisons. From the table below, we found that **the performance improvements are statistical significant**.
>
> **Table 1: Comparison of different decoding strategies on POPE.**
>
> | **Method**  | **POPE \(R)** |    | **POPE \(P)** |   | **POPE \(A)** |    |
> |-------------|--------------|-----------|--------------|---------|--------------|---------|
> | ****        | Accuracy↑     | F1 Score↑   | Accuracy↑     | F1 Score↑ | Accuracy↑     | F1 Score↑ |
> | **Default** | 83.86        | 82.68     | 80.82        | 79.54   | 76.42        | 76.61   |
> | **OPERA**   | 88.85        | 88.67     | 82.77        | 83.4    | 79.16        | 80.93   |
> | **VCD**     | 87.2         | 87.17     | 83.08        | 83.07   | 77.7         | 79.14   |
> | **ICD**    | 83.15        | 83.91     | 83.15        | 83.91   | 79.13        | 80.41   |
> | **SID**   |89.53 ± 0.15|	89.31 ± 0.18|	85.13 ± 0.15	|86.03 ± 0.19|	83.14 ± 0.22|	82.18 ± 0.09|
> |**ATED#**|89.89 ± 0.11（p<0.05）|	89.06 ± 0.19(p=0.07)|	86.51 ± 0.21（p<0.05）|	85.49 ± 0.08（p<0.05）|	82.84 ± 0.15（p<0.05）|	82.34 ± 0.07（p<0.05）	|

---

> ### Author Response · Authors · 2025-11-22
> **Response to Reviewer oHTM (part 2/3)**
>
> > W3. While ATED claims to be “training-free,” the computation overhead of multi-model forward passes is non-trivial; the paper does not provide detailed latency or cost comparisons under identical hardware constraints.
>
> **Answer:** We thank the reviewer for the valuable suggestion and totally agree that deployment efficiency is crucial for the practical applicability of ATED.
>
>
> We further conduct a latency evaluation and report the results in **Table 2 of this rebuttal.**, where we compare three approaches standard decoding (default), the ensemble decoding baseline ED[1], and ATED variants on three metrics, CHAIR_S (with the max new token length set to 512), the average caption length, and the per-image inference latency. ATED achieves better hallucination-related performance while maintaining a similar order of magnitude in inference latency as ED, thus offering a **more favorable performance–efficiency trade-off**.
>
> Regarding GPU memory, the proposed ATED can be accomondated on a single NVIDIA L20 (48 GB) GPU, with all components fully loaded. In Section 3.7 of the revised manuscript, we have added a dedicated efficiency analysis to clarify this deployment setup and to emphasize that ATED can still be deployed on a single GPU while providing substantially stronger hallucination mitigation than ED. Therefore, in scenarios where additional training or large GPU clusters are costly or impractical, ATED remains a **practically feasible** and **lightweight ensemble decoding strategy**.
>
> **Table 2: Inference latency and performance results.**
>
> | **Method** |   **s** |**ε** | **CHAIR_S (512)↓**  | **Average Length↑** | **Inference Latency↓** |
> |------------|-------|-------|------------------|----------------------|-----------------------|
> | default    |  -     |  -     |       51.3           |     100.6                 |   **6.4**                 |
> | **ED**         |   -    |  -     |      43.0            |   100.1                   |                16.4       |
> | **ATED\#**   |  0.5   |2.5    |      $\underline{40.4}$          |     $\underline{103.6}$                |    $\underline{12.3}$                  |
> | **ATED\#**      |  0.2    |  1.5   |  **36.2**           |  **105.2**                    |           16.5       |
>
> [1]Cho, Y., Kim, K., Hwang, T., & Cho, S. (2025). Do You Keep an Eye on What I Ask? Mitigating Multimodal Hallucination via Attention-Guided Ensemble Decoding. arXiv preprint arXiv:2505.17529.
>
> &nbsp;
> > Q1. How many LVLMs were actually ensembled in each experiment, and how does performance scale with the number of models (N = 2 → 3 → 4)?
>
> **Answer:** We thank the reviewer for the valuable comment. Here we further provide the performance scaling of ATED with respect to ensemble size. As shown in **Table 3 below**, when the ensemble size grows from **N=2 to N=4**, the metirc **Accuracy on the POPE\(R) benchmark improves consistently**, indicating that, with appropriately chosen model combinations, ensembling can effectively exploit the complementarity among models.
>
> **Table 3: LVLMs performance scale with different Ensemble size on POPE \(R).**
> | Ensemble size (N) | Accuracy↑|  F1 Score↑ |
> |-------------------|------------------|-------------------|
> | N = 2             | 89.39            |$\underline{89.39}$   |
> |N = 3              |   $\underline{89.83}$    |     89.35       |
> |N = 4              |    **90.01**	   |   **89.68**     |
>
> Here,
>
> **N = 2** denotes the ensemble of LLaVA-1.5 and InstructBLIP;
>
> **N = 3** denotes the ensemble of LLaVA-1.5, InstructBLIP, and LLaVA-NeXT;
>
> **N = 4** denotes the ensemble of LLaVA-1.5, InstructBLIP, LLaVA-NeXT and LLaVA-1.5 based SID.

---

> ### Author Response · Authors · 2025-11-22
> **Response to Reviewer oHTM (part 3/3)**
>
> > Q2. How sensitive is ATED to the hyperparameters?
> >
> **Answer:** We thank the reviewer for this question. We have performed ablation studies on the **hyperparameters noise steps T and** the **hyperparameter a that controls the intensity of visual contrastive decoding**. The detailed results are reported in **Appendix E.3** “IMPACT OF VISION PERTURBATIONS” (**Tables 7 and 8**). For convenience, the results are further **summarized in Tables 4 and 5 of this rebuttal**.
>
>
> **Table 4: Evaluation on POPE（R）and MME with varying noise steps T.**
>
> | Noise step | POPE\(R) (Accuracy↑) | POPE\(R) (F1 Score↑) | MME subset (Accuracy+ ↑) |
> |-----------|----------------------|----------------------|---------------------------|
> | 0   | 85.90         | 84.19         | $\underline{616.67}$         |
> | 200 | $\underline{87.04}$ | $\underline{85.86}$ | **636.67**     |
> | 500 | 86.73         | 85.44         | 608.33  |
> | 700 | 86.88         | 85.49         | 591.67  |
> | 999 | **87.17**     | **86.59**     | 576.67  |
>
> **Table 5: Evaluation on MME subset with varying α.**
>
>
> | α   | Existence↑        | Count↑            | Position↑           | Color↑              | Total Accuracy+↑      |
> |-----|-------------------|-------------------|---------------------|---------------------|------------------------|
> | 0.5 | $\underline{180.00}$ | $\underline{143.33}$ | 131.67              | $\underline{140}$  | 595.00                |
> | 0.7 | $\underline{180.00}$ | $\underline{143.33}$ | $\underline{136.67}$ | $\underline{140}$  | $\underline{600.00}$  |
> | 1.0 | **185.00**        | **158.33**        | **138.33**          | **155**             | **636.67**            |
>
> &nbsp;
> > Q3. Can the uncertainty-guided weighting be estimated within a single model using internal heads or adapters, rather than across multiple LVLMs?
>
> **Answer:** Conceptually, it is indeed possible to “simulate” an ensemble within a single LVLM by adding multiple heads or adapters. However, this approach is fundamentally different from the design of ATED and is non-trivial in practice. ATED relies on token-level uncertainty defined over the full vocabulary distribution, whereas in a standard LVLM architecture only the final output layer corresponds to a well-calibrated vocabulary-level probability distribution. Intermediate layers (including multi-head attention blocks or adapters) operate purely in feature space and do not directly produce a normalized distribution over the vocabulary.
>
> To obtain usable uncertainty estimates from internal heads, one would need to attach additional output projections to the full vocabulary at multiple intermediate layers and then train or fine-tune these heads so that they can produce valid next-token distributions. This is effectively equivalent to redesigning a new, multi-head, training-based single-model architecture, rather than the training-free ensemble of multiple independently trained LVLMs that we propose, and it would also introduce substantial training cost and architectural complexity.

---

> ### Author Response · Authors · 2025-11-26
> **Kindly Reminder**
>
> Dear Reviewer oHTM,
>
> We deeply appreciate your valuable feedback and the time you've taken to review our work, especially during this busy period.
>
> We are reaching out to kindly inquire about the current status of your review regarding our submission. We have uploaded the revised manuscript with updates **highlighted in crimson blue**, and updated corresponding details in the rebuttal to reflect the current version. Based on your comments, we have conducted the following additional experiments and analyses:
>
> 1. Clarification of the **conceptual differences** between ATED and ED/VCD, along with an analysis of **UGO’s contribution**;
> 2. Statistical **significance tests** and reporting of **variance**;
> 3. An analysis of **deployment efficiency**;
> 4. An analysis of **performance scaling with the number of ensemble models**;
> 5. Detailed explanation of the **hyperparameter experiments**;
> 6. Clarification that **single-model internal heads or adapters** cannot currently replace multi-model uncertainty weighting.
>
> We sincerely hope that our responses have adequately addressed your concerns. Furthermore, we are eager to address any additional queries you might have, which will enable us to enhance our work further.
>
> Once again, thank you for your guidance and support.
>
> Best, Authors of Paper 8276

---

> > ### Comment · Reviewer_oHTM · 2025-11-27
> >
> > Thank you for running the additional experiments and providing such detailed explanations. The updated results really helped clear up most of my questions, and I appreciate the extra work you put into these points. However, I still feel that the novelty of the proposed method is not sufficiently strong, so I will keep my original score.

---

> > > ### Author Response · Authors · 2025-12-04
> > >
> > > We sincerely appreciate your positive feedback. We are glad that our response has addressed most of your concerns, and we have incorporated your comments in the latest revision of the manuscript.

---

### Official Review · Reviewer_5jsQ · 2025-10-29

**Soundness:** 2
**Presentation:** 2
**Contribution:** 2
**Rating:** 4
**Confidence:** 4

**Summary:**

The paper proposes Adaptive Token Ensemble Decoding (ATED), a training-free, token-level ensemble method to curb object hallucination in LVLMs. ATED aggregates per-step predictions from multiple LVLMs using uncertainty-adaptive weights and fuses diverse decoding paths to strengthen grounding and semantic consistency—without extra training or model-specific modules. On standard hallucination benchmarks, ATED lowers hallucination rates while preserving fluency and task relevance, outperforming prior objectives and post-hoc detectors.

**Strengths:**

1. The topic is interesting and tries to address an important problem.

2. The paper writing is easy to follow.

**Weaknesses:**

1. Beyond entropy, can uncertainty be measured with alternative metrics?

2. I’m unclear on the exact decoding procedure. After computing model-specific uncertainty weights, which model (or aggregation) actually drives decoding? Does this operate token-by-token only, or can it decode full sentences? If full sentences, must outputs from all models be forced to match exactly?

3. Please provide efficiency measurements for the entire procedure.

4. I would like deeper analysis—for example, showing whether higher-performing models systematically receive larger weights.

5. Can the proposed method be applied to the Qwen-2.5-VL family?

6. For the benchmark discussion, note that several recent studies [1, 2, 3] address both hallucination and maintain performance (even some improvement) on general scenario. I recommend the authors add some benchmarks like OCRBench, MMMU, MME etc.

[1] Mitigating Object Hallucinations via Sentence-Level Early Intervention.

[2] A topic-level self-correctional approach to mitigate hallucinations in mllms.

[3] Rlaif-v: Aligning mllms through open-source ai feedback for super gpt-4v trustworthiness.

**Questions:**

See above.

---

> ### Author Response · Authors · 2025-11-22
> **Response to Reviewer 5jsQ (part 1/3)**
>
> We greatly appreciate the reviewer’s valuable and insightful feedback. Below, we have provided detailed responses to each of your questions.
>
> > W1. Beyond entropy, can uncertainty be measured with alternative metrics?
> >
> **Answer:** We thank the reviewer for this valuable suggestion. In the Adaptive Uncertainty-Guided Weight subsection of our paper, we have already included a confidence-based weighting variant that uses the Maximum Softmax Probability as an alternative uncertainty signal, and the corresponding results are reported in **Table 3 (the row “Confidence-based”) of the paper**. More detailed results are presented in Table 1 below. As shown there, our **ATED** method consistently **outperforms this confidence-based variant across all three settings on the POPE MSCOCO benchmark.**
>
> **Table 1: Results comparison on the POPE MSCOCO benchmark.**
>
> | **Setting** | **Method**       | **Accuracy** | **F1 Score** |
> |-------------|------------------|--------------|--------------|
> | **Random**      | Confidence-based | 88.13        | 86.76        |
> |             | ATED             | **88.97**        | **88.29**       |
> | **Popular**     | Confidence-based | 86.27        | 84.79        |
> |             | ATED             | **87.57**        | 86.58        |
> | **Adversarial** | Confidence-based | 84.77        | 83.4         |
> |             | ATED             | **85.37**        | **84.64**        |
>
> &nbsp;
> > W2. I’m unclear on the exact decoding procedure. After computing model-specific uncertainty weights, which model (or aggregation) actually drives decoding? Does this operate token-by-token only, or can it decode full sentences? If full sentences, must outputs from all models be forced to match exactly?>
>
> **Answer:** After computing model-specific uncertainty weights, decoding is driven by the aggregated vocabulary distribution. At each decoding step $t$, we assume that a shared prefix $x_{0:t-1} = (x_0, x_1, \dots, x_{t-1})$ has already been generated. Given this prefix and the visual input, each ensembled LVLM $M_m$ independently produces a vocabulary-level distribution $d_t^{(m)}$ over the next token $x_t$ (via its logits). We then compute an uncertainty score for each model at step $t$, convert these scores into normalized weights $\{\lambda_t^{(m)}\} \in [0,1]$, and form the aggregated distribution: $d_t = \sum_m \lambda_t^{(m)} d_t^{(m)}$.
>
> The next token $x_t$ is selected from this fused distribution $d_t$, appended to the prefix to obtain $x_{0:t}$, and the updated prefix is fed back to all models for the subsequent step. Consequently, ATED operates strictly in a **token-by-token fashion** and thus does not require alignment at the sentence level.
>
> &nbsp;
> > W3. Please provide efficiency measurements for the entire procedure.
> >
> **Answer:** We thank the reviewer for the valuable suggestion and totally agree that deployment efficiency is crucial for the practical applicability of ATED.
>
> We further conduct a efficiency evaluation and report the results in **Table 2 of this rebuttal.**, where we compare three approaches standard decoding (default), the ensemble decoding baseline ED[1], and ATED variants on three metrics, CHAIR_S (with the max new token length set to 512), the average caption length, and the per-image inference latency. ATED achieves better hallucination-related performance while maintaining a similar order of magnitude in inference latency as ED, thus offering a more favorable performance–efficiency trade-off.
>
> Regarding GPU memory, the proposed ATED can be accomondated on a single NVIDIA L20 (48 GB) GPU, with all components fully loaded. In Section 3.7 of the revised manuscript, we have added a dedicated efficiency analysis to clarify this deployment setup and to emphasize that ATED can still be deployed on a single GPU while providing substantially stronger hallucination mitigation than ED. Therefore, in scenarios where additional training or large GPU clusters are costly or impractical, ATED remains a practically feasible and lightweight ensemble decoding strategy.
>
> **Table 2: Inference latency and performance results.**
>
> | **Method** |   **s** |**ε** | **CHAIR_S (512)↓**  | **Average Length↑** | **Inference Latency↓** |
> |------------|-------|-------|------------------|----------------------|-----------------------|
> | default    |  -     |  -     |       51.3           |     100.6                 |   **6.4**                 |
> | **ED**         |   -    |  -     |      43.0            |   100.1                   |                16.4       |
> | **ATED\#**   |  0.5   |2.5    |      $\underline{40.4}$         |     $\underline{103.6}$              |    $\underline{12.3}$                  |
> | **ATED\#**      |  0.2    |  1.5   |  **36.2**           |  **105.2**                    |           16.5       |
>
> [1]Cho, Y., Kim, K., Hwang, T., & Cho, S. (2025). Do You Keep an Eye on What I Ask? Mitigating Multimodal Hallucination via Attention-Guided Ensemble Decoding. arXiv preprint arXiv:2505.17529.

---

> ### Author Response · Authors · 2025-11-22
> **Response to Reviewer 5jsQ (part 2/3)**
>
> > W4. I would like deeper analysis—for example, showing whether higher-performing models systematically receive larger weights.
> >
> **Answer:**  We agree that understanding how ATED allocates weights across different models is crucial for interpreting the behavior of the ensemble. To this end, we have added a quantitative analysis on **POPE and MME**. For each benchmark, we first identify the strongest single LVLM (e.g., **LLaVA-NeXT on POPE, and LLaVA-1.5 outperforming InstructBLIP on MME**), then run **ATED& and ATED#** on the full evaluation set and **collect the weight allocation statistics at each decoding step**. The results show that, on POPE, the strongest model receives the highest weight in approximately **75%** of decoding steps, and this proportion is about **82%** on the MME subset. This indicates that our uncertainty-guided mechanism indeed tends to favor the stronger model overall, while still dynamically incorporating useful information from other models when they are more confident about specific local tokens.
>
> &nbsp;
>
> > W5. Can the proposed method be applied to the Qwen-2.5-VL family?
> >
> **Answer:** Following your excellent advice, we applied the proposed ATED to the QWen series models. Specifically, we selected two VL models, Qwen2.5-VL-3B and Qwen2.5-VL-7B, and conducted the experiments separately. We then used ATED to aggregate these two models. As shown in Table 3 of this response, the proposed ATED outperforms both models, demonstrating its scalability. We further evaluated ATED on Qwen2.5-VL 3B and 7B within the Qwen-VL family, and the results are shown in Table 3 below. The improvements on MMMU overall indicate that the method has broad applicability.
>
> **Table 3: MMMU evaluation results on Qwen-2.5-VL family.**
> | **Model**     | **MMMU (val) overall** |
> |---------------|----------------|
> | Qwen2.5-VL-3B | 51.82          |
> | Qwen2.5-VL-7B | 57.50          |
> | **ATED**      | **59.22**   |
>
> [1]Yao Y, Wu H, Liu M, et al. Determine-then-ensemble: Necessity of top-k union for large language model ensembling[J]. arXiv preprint arXiv:2410.03777, 2024.

---

> ### Author Response · Authors · 2025-11-22
> **Response to Reviewer 5jsQ (part 3/3)**
>
> > W6. For the benchmark discussion, note that several recent studies [1, 2, 3] address both hallucination and maintain performance (even some improvement) on general scenario. I recommend the authors add some benchmarks like OCRBench, MMMU, MME etc.
> >
> **Answer:** Thank you for the suggestion to include broader benchmarks. **For clarity, Section 3.5 (Figure 3) of our paper already reports MME subset results**. In particular, we conduct a systematic and comprehensive evaluation of ATED on the hallucination subset of MME. The corresponding results are now reported in Table 4 below. Overall, ATED achieves substantial gains across four tasks, outperforming the default LVLM and other baselines; on the accuracy+ metric the improvements are at least +61.7% and +54.2%, respectively.
>
> We also agree with the reviewer’s point that hallucination mitigation should be reported together with general capability. To this end, we additionally evaluate on **MMMU using the Validation subset (900 images)**. The experimental results reported in Table 5 below show that our methods (**ATED\*, ATED&, and ATED\#**) achieve performance that is comparable to or even better than the strong **SID baseline**. Notably, **ATED# attains the highest score of 35.9** among all models and decoding strategies, clearly outperforming LLaVA-1.5 and InstructBLIP under **default, VCD, and SID settings**. Moreover, ATED& and ATED* both exceed the two baseline models under their default settings and, in some cases, match or surpass advanced decoding strategies such as SID. These results show that **ATED reduces hallucinations without sacrificing general multimodal reasoning performance**, and in some instances improves it.
>
> **Table 4: MME evaluation results on different decoding strategies on LLaVA-1.5 and InstructBLIP.**
> | Model    | Method    | Existence↑ | Count↑ | Position↑             | Color↑                | Total   | Model        | Existence↑             | Count↑ | Position↑             | Color↑                | Total   |
> |---------|-----------|------------|--------|------------------------|-----------------------|---------|--------------|------------------------|--------|------------------------|-----------------------|---------|
> | LLaVA-1.5 | Default | 175.0      | 125.0  | $\underline{120.0}$   | $\underline{155.0}$   | 575.0   | InstructBLIP | $\underline{170.0}$    | 78.8   | 65.5                   | 106.7                 | 421.0   |
> |          | OPERA   | 174.7      | 115.7  | 110.7                  | 141.0                 | 542.0   |              | 158.3                  | 81.3   | 62.9                   | 109.7                 | 412.1   |
> |          | VCD     | 174.4      | 124.1  | 119.4                  | 153.6                 | 571.5   |              | 166.7                  | 82.1   | 67.1                   | $\underline{128.0}$   | $\underline{443.8}$ |
> |          | ICD     | $\underline{185.0}$ | 117.9  | 117.5                  | **162.1**             | $\underline{582.5}$ |              | 166.7                  | 80.8   | $\underline{67.9}$     | 118.3                 | 433.8   |
> |          | SID     | 182.0      | $\underline{127.0}$ | 116.0                  | 139.0                 | 564.0   |              | 165.0                  | $\underline{89.2}$     | 65.9                   | 114.3                 | 434.4   |
> |          | **ATED#** | **185.0** | **158.3** | **138.3**             | $\underline{155.0}$   | **636.7** |              | **185.0**              | **158.3** | **138.3**             | **155.0**             | **636.7** |
>
> **Table 5: MMMU evaluation results on different decoding strategies.**
>
> | Model        | Method  | Validation Overall ↑ |
> |:------------:|:-------:|:------------------:|
> | LLaVA-1.5    | Default | 33.3               |
> |              | VCD     | 34.1               |
> |              | SID     | $\underline{34.4}$        |
> | InstructBLIP | Default | 32.9               |
> |              | VCD     | 33.4               |
> |              | SID     | 34.1               |
> | **ATED\***   | -       | 33.9           |
> | **ATED\&**   | -       |34.3         |
> | **ATED\#**   | -      | **35.9**           |
>
>
> Here,
>
> **ATED*** denotes our ensemble method of LLaVA-1.5 and InstructBLIP without vision contrastive decoding;
>
> **ATED&** denotes the ensemble of LLaVA-1.5 and InstructBLIP;
>
> **ATED#** denotes the ensemble of LLaVA-1.5, InstructBLIP, and LLaVA-NeXT.

---

> ### Author Response · Authors · 2025-11-26
> **Kindly Reminder**
>
> Dear Reviewer 5jsQ,
>
> We deeply appreciate your valuable feedback and the time you've taken to review our work, especially during this busy period.
>
> We are reaching out to kindly inquire about the current status of your review regarding our submission. We have uploaded the revised manuscript with updates **highlighted in crimson blue**, and updated corresponding details in the rebuttal to reflect the current version. Based on your comments, we have conducted the following additional experiments and analyses:
>
> 1. Experiments with **uncertainty measures beyond entropy**;
> 2. A more detailed description of the **decoding procedure**;
> 3. An analysis of **deployment efficiency**;
> 4. Deeper analysis  of **model weight allocation during decoding**;
> 5. Discussion and experiments on the **Qwen2.5-VL series**;
> 6. Additional evaluations on **broader benchmarks**.
>
> We sincerely hope that our responses have adequately addressed your concerns. Furthermore, we are eager to address any additional queries you might have, which will enable us to enhance our work further.
>
> Once again, thank you for your guidance and support.
>
> Best, Authors of Paper 8276

---

### Official Review · Reviewer_akFE · 2025-11-01

**Soundness:** 3
**Presentation:** 3
**Contribution:** 3
**Rating:** 6
**Confidence:** 4

**Summary:**

The paper proposes Adaptive Token Ensemble Decoding (ATED), a training‑free framework that mitigates hallucinations in large vision‑language models by fusing next‑token logits from multiple LVLMs with uncertainty‑guided weights. ATED generates perturbed image variants, applies contrastive decoding, and greedily minimizes entropy to assign importance weights to each model, allowing a trade‑off between inference latency and accuracy. Experiments on POPE, CHAIR and MME demonstrate consistent gains over individual backbones and several plug‑and‑play decoding baselines, with ablations analysing weighting strategies and latency.

**Strengths:**

(1) Training-free, plug-and-play method that leverages existing LVLMs without retraining; works across several backbones.
(2) Consistent empirical gains on POPE / CHAIR / MME over strong decoding baselines (VCD, ICD, SID).
(3) Ablations + latency knob make the method well-diagnosed and practically tunable.

**Weaknesses:**

(1) The paper compares a multi-model ensemble to single-model baselines; real-world feasibility of running 2–3 LVLMs + perturbations per token is unclear.

(2) The paper also lacks comparison to 2025 ED / FastED / iTaD / IFCD-style plug-and-play hallucination mitigators, weakening the “significantly outperforms SOTA” claim.

**Questions:**

(1) Can you provide a fair multi-model baseline (e.g., 3 LVLMs with uniform logit averaging, same perturbations) to isolate the gain from your uncertainty-greedy weighting?

(2) Can you add or report results vs 2025 ED/FastED-type methods on at least one of POPE/CHAIR to strengthen the SOTA claim?

---

> ### Author Response · Authors · 2025-11-22
> **Response to Reviewer akFE (part 1/2）**
>
> We greatly appreciate the reviewer’s valuable and insightful feedback. Below, we have provided detailed responses to each of your questions.
>
> > W1. The paper compares a multi-model ensemble to single-model baselines; real-world feasibility of running 2–3 LVLMs + perturbations per token is unclear.
> >
>
> **Answer:** We thank the reviewer for the valuable suggestion and totally agree that deployment efficiency is crucial for the practical applicability of ATED.
>
> We conduct a latency evaluation and report the results in **Table 1 of this response**, where we compare three approaches standard decoding (default), the ensemble decoding baseline ED[1], and ATED variants on three metrics, CHAIR_S (with the max new token length set to 512), the average caption length, and the per-image inference latency. ATED achieves better hallucination-related performance while maintaining a similar order of magnitude in inference latency as ED, thus offering a more **favorable performance–efficiency trade-off**.
>
> Regarding GPU memory, the proposed ATED can be accomondated on a single NVIDIA L20 (48 GB) GPU, with all components fully loaded. In Section 3.7 of the revised manuscript, we have added a dedicated efficiency analysis to clarify this deployment setup and to emphasize that ATED can still be deployed on a single GPU while providing substantially stronger hallucination mitigation than ED. Therefore, in scenarios where additional training or large GPU clusters are costly or impractical, ATED remains a **practically feasible** and **lightweight ensemble decoding strategy**.
>
>
> **Table 1: Inference latency and performance results.**
>
> | **Method** |   **s** |**ε** | **CHAIR_S (512)↓**  | **Average Length↑** | **Inference Latency↓** |
> |------------|-------|-------|------------------|----------------------|-----------------------|
> | default    |  -     |  -     |       51.3           |     100.6                 |   **6.4**                 |
> | **ED**         |   -    |  -     |      43.0            |   100.1                   |                16.4       |
> | **ATED\#**   |  0.5   |2.5    |      $\underline{40.4}$          |     $\underline{103.6}$                |    $\underline{12.3}$                  |
> | **ATED\#**      |  0.2    |  1.5   |  **36.2**           |  **105.2**                    |           16.5       |
>
> >W2. The paper also lacks comparison to 2025 ED / FastED / iTaD / IFCD-style plug-and-play hallucination mitigators, weakening the “significantly outperforms SOTA” claim.
>
> **Answer:** For the hallucination mitigator ED[1], in **Appendix Table 5**, we have already reported the **CHAIR** evaluation results of ED under different decoding strategies (**see the Table 2 below**). Under a unified setting of **max_new_tokens=512**, our method consistently outperforms all baselines on the** CHAIR_S** metric, demonstrating stronger robustness and adaptability across different generation lengths. In addition, as shown **in Table 1 of this response**, the latency of the ATED is similar to or even better than ED.
>
> **Table 2: CHAIR evaluation results on different decoding strategies.**
>
> | **Model** | **Method** | **CHAIR_S ↓** | **CHAIR_I ↓** |
> |-----------|------------|-------------|------------|
> | LLaVA-1.5 | Default    | 51.3        | 16.8       |
> |           | OPERA      | 46.4        | $\underline{13.0}$         |
> |           | VCD        | 51.7        | 15.6       |
> |           | ICD        | 47.4        | 13.9       |
> |           | SID        | 44.2        | **12.2**      |
> |           | **ED**         | $\underline{43.0}$       | 14.0      |
> | **ATED#**     | -          | **34.0**          | 17.1       |
>
> [1]Cho, Y., Kim, K., Hwang, T., & Cho, S. (2025). Do You Keep an Eye on What I Ask? Mitigating Multimodal Hallucination via Attention-Guided Ensemble Decoding. arXiv preprint arXiv:2505.17529.

---

> ### Author Response · Authors · 2025-11-22
> **Response to Reviewer akFE (part 2/2）**
>
> > Q1. Can you provide a fair multi-model baseline (e.g., 3 LVLMs with uniform logit averaging, same perturbations) to isolate the gain from your uncertainty-greedy weighting?
> >
>
> **Answer:** In Section 3.6, under the subsection “Adaptive Uncertainty-Guided Weight,” we have already provided a multi-model ensemble baseline (**the row “Uniform” in Table 3 of the paper**). Specifically, on the **MSCOCO subset of the POPE benchmark**, we construct an ensemble of three LVLMs (**LLaVA-1.5, InstructBLIP, and LLaVA-NeXT**), using exactly the same perturbation settings as ATED, but applying only uniform logit averaging, without our proposed uncertainty-greedy weighting (UGO).
>
> The experimental results demonstrate that our ATED variant achieves consistent improvements over this uniform ensemble. For example, the Accuracy improves from (**84.57 vs 87.57**), and the F1 Score from (**84.37 vs 86.58**).
>
> &nbsp;
> >Q2. Can you add or report results vs 2025 ED/FastED-type methods on at least one of POPE/CHAIR to strengthen the SOTA claim?
> >
> **Answer:** We thank the reviewer for this insightful suggestion. Following your comments, we conducted a systematic comparison between ATED and ED[1] on the **POPE benchmark** under the same experimental setup as in the original ED paper. Overall, ATED achieves an **average Accuracy** of **86.50**, which is higher than **ED’s 86.31**; in terms of the **average F1 Score**, ATED also outperforms ED (**86.03 vs. 85.86**).

---

> ### Author Response · Authors · 2025-11-26
> **Kindly Reminder**
>
> Dear Reviewer akFE,
>
> We deeply appreciate your valuable feedback and the time you've taken to review our work, especially during this busy period.
>
> We are reaching out to kindly inquire about the current status of your review regarding our submission. We have uploaded the revised manuscript with updates **highlighted in crimson blue**, and updated corresponding details in the rebuttal to reflect the current version. Based on your comments, we conducted the following additional experiments and analyses:
>
> 1. An analysis of **deployment efficiency**;
> 2. Comparison with the **2025 ED** plug-and-play hallucination mitigators;
> 3. Clarification regarding the “**missing fair ensemble baseline**” experiments;
> 4. Comparison with 2025 ED on the **POPE** and **CHAIR benchmarks**.
>
> We sincerely hope that our responses have adequately addressed your concerns. Furthermore, we are eager to address any additional queries you might have, which will enable us to enhance our work further.
>
> Once again, thank you for your guidance and support.
>
> Best, Authors of Paper 8276

---

### Official Review · Reviewer_caNz · 2025-11-01

**Soundness:** 3
**Presentation:** 3
**Contribution:** 2
**Rating:** 4
**Confidence:** 3

**Summary:**

The paper proposes Adaptive Token Ensemble Decoding (ATED), a training-free ensemble framework designed to mitigate multimodal hallucinations in large vision-language models (LVLMs). ATED dynamically fuses token-level logits from multiple LVLMs based on uncertainty-guided weighting, allowing it to leverage the complementary strengths of each model during inference. The method is evaluated on several benchmarks including POPE, CHAIR, and MME, and achieves significant improvements in hallucination reduction without requiring retraining or fine-tuning. The authors further discuss the trade-off between inference latency and accuracy and analyze various ensemble strategies.

**Strengths:**

1. The paper introduces a clear and well-motivated idea: ensemble decoding at the token level across multiple LVLMs guided by adaptive uncertainty. This fine-grained approach extends ensemble learning into multimodal generation, which is both innovative and practically relevant.
2. ATED does not require additional training, making it broadly applicable across existing LVLMs and compatible with open-source backbones like LLaVA, InstructBLIP, and MiniGPT-4.
3. The paper includes comparisons on multiple benchmarks (POPE, CHAIR, MME) and provides ablation studies showing the contribution of each component, such as uncertainty-guided weighting and greedy optimization.

**Weaknesses:**

1. The proposed ATED framework requires simultaneous inference across multiple LVLMs, which substantially increases GPU memory usage and deployment cost. Figure 4 also shows that inference latency can increase up to six times compared to standard decoding. In contrast, other training-free approaches such as VCD typically introduce at most a twofold increase in latency. This raises concerns about ATED’s scalability and practicality in real-world applications where efficiency is critical.

2. The experiments mainly compare ATED with training-free decoding strategies (e.g., VCD, ICD, SID, OPERA), but do not include training-based hallucination mitigation methods, such as instruction-tuning or preference optimization. Including such comparisons would better demonstrate ATED’s effectiveness and contributions.

3. While the paper reports improvements on hallucination-related metrics such as POPE, CHAIR, and MME (hallucination subset), it lacks qualitative and quantitative evaluation of overall generation quality (e.g., fluency, coherence, descriptive richness). For instance, benchmarks like RefoMB, LLaVA-Bench, MMStar, and MM-Vet could assess how ATED affects long-form captioning and reasoning under complex visual conditions. Without these results, it remains unclear whether ATED preserves or degrades naturalness in extended outputs.

4. The paper focuses mainly on hallucination benchmarks but does not discuss whether ATED affects general-purpose multimodal understanding. Evaluations on textVQA, DocVQA, InfoVQA, and VQAv2 would help determine whether the ensemble decoding alters the model’s broader reasoning or comprehension abilities. It is important to verify that the hallucination reduction does not come at the cost of decreased general accuracy or robustness on standard multimodal benchmarks.

**Questions:**

1. Include an analysis of deployment efficiency, especially GPU memory consumption and throughput, to clarify ATED’s practical applicability.
2. Add comparisons with training-based hallucination mitigation methods, such as fine-tuned models using preference alignment or reinforcement learning.
3. Extend the evaluation to long-form generation benchmarks and provide both evaluations of fluency and relevance.
4. Evaluate ATED’s impact on general-purpose performance using standard multimodal benchmarks (e.g., textVQA, DocVQA, VQAv2).

---

> ### Author Response · Authors · 2025-11-22
> **Response to Reviewer caNz（part 1/3）**
>
> We greatly appreciate the reviewer’s valuable and insightful feedback. Below, we have provided detailed responses to each of your questions.
>
> > W1. The proposed ATED framework requires simultaneous inference across multiple LVLMs, which substantially increases GPU memory usage and deployment cost. Figure 4 also shows that inference latency can increase up to six times compared to standard decoding. In contrast, other training-free approaches such as VCD typically introduce at most a twofold increase in latency. This raises concerns about ATED’s scalability and practicality in real-world applications where efficiency is critical.
> >
>
> **Answer:** We thank the reviewer for the valuable suggestion and totally agree that deployment efficiency is crucial for the practical applicability of ATED. We first clarify the interpretation of Figure 4: at the fastest operating point of ATED (approximately **12.5s** with **s=0.5,ε=2.5**), the latency is about **2 times** that of standard decoding (the **“default” node** in Figure 4, **6.4s**), rather than “**6 times**.” This misunderstanding mainly arises because the y-axis in Figure 4 starts from 5 instead of 0, which visually exaggerates the relative difference.
>
> We further conduct a latency evaluation and report the results in **Table 1 of this response.**, where we compare three approaches standard decoding (default), the ensemble decoding baseline ED[1], and ATED variants on three metrics, CHAIR_S (with the max new token length set to 512), the average caption length, and the per-image inference latency. ATED achieves better hallucination-related performance while maintaining a similar order of magnitude in inference latency as ED, thus offering a more favorable performance–efficiency trade-off.
>
> Regarding GPU memory, the proposed ATED can be accommodated on a single NVIDIA L20 (48 GB) GPU, with all components fully loaded. In Section 3.7 of the revised manuscript, we have added a dedicated efficiency analysis to clarify this deployment setup and to emphasize that ATED can still be deployed on a single GPU while providing substantially stronger hallucination mitigation than ED. Therefore, in scenarios where additional training or large GPU clusters are costly or impractical, ATED remains a practically feasible and lightweight ensemble decoding strategy.
>
>
> **Table 1: Inference latency and performance results.**
>
> | **Method** |   **s** |**ε** | **CHAIR_S (512)↓**  | **Average Length↑** | **Inference Latency↓** |
> |------------|-------|-------|------------------|----------------------|-----------------------|
> | default    |  -     |  -     |       51.3           |     100.6                 |   **6.4**                 |
> | **ED**         |   -    |  -     |      43.0            |   100.1                   |                16.4       |
> | **ATED\#**   |  0.5   |2.5    |       $\underline{40.4}$           |     $\underline{103.6}$                |    $\underline{12.3}$                 |
> | **ATED\#**      |  0.2    |  1.5   |  **36.2**           |  **105.2**                    |           16.5       |
>
>
> [1] Cho, Y., Kim, K., Hwang, T., & Cho, S. (2025). Do You Keep an Eye on What I Ask? Mitigating Multimodal Hallucination via Attention-Guided Ensemble Decoding. arXiv preprint arXiv:2505.17529.

---

> > ### Author Response · Authors · 2025-11-26
> > **Kindly Reminder**
> >
> > Dear Reviewer caNz,
> >
> > We deeply appreciate your valuable feedback and the time you've taken to review our work, especially during this busy period.
> >
> > We are reaching out to kindly inquire about the current status of your review regarding our submission. We have uploaded the revised manuscript with updates **highlighted in crimson blue**, and updated corresponding details in the rebuttal to reflect the current version. Based on your comments, we conducted additional experiments:
> >
> > 1. An analysis of **deployment efficiency**;
> > 2. Comparative experiments with **training-based hallucination mitigation methods**;
> > 3. Extended evaluation on **long-form generation benchmarks**;
> > 4. Additional evaluation of general-purpose performance on **standard multimodal benchmarks**.
> >
> > We sincerely hope that our responses have adequately addressed your concerns. Furthermore, we are eager to address any additional queries you might have, which will enable us to enhance our work further.
> >
> > Once again, thank you for your guidance and support.
> >
> > Best, Authors of Paper 8276

---

> ### Author Response · Authors · 2025-11-22
> **Response to Reviewer caNz（part 2/3）**
>
> > W2. The experiments mainly compare ATED with training-free decoding strategies (e.g., VCD, ICD, SID, OPERA), but do not include training-based hallucination mitigation methods, such as instruction-tuning or preference optimization. Including such comparisons would better demonstrate ATED’s effectiveness and contributions.
> >
>
> **Answer:** Following your insightful suggestion, we have added comparisons with **RLAIF-V** as a representative training-based hallucination mitigation method[1]. RLAIF-V adopts a “Feedback From Peer” strategy and is trained via preference optimization to reduce hallucinations in multimodal large language models. To keep our setting consistent with the main experiments, we use the LLaVA based RLAIF-V model for evaluation. More specifically, we assess the RLAIF-V model on the POPE, MME, and MM-Vet benchmarks. The corresponding results are summarized in **Table 2 of this response**. Over hallucination-related metrics across all three benchmarks, **the proposed ATED variants outperform RLAIF-V**. At the same time, ATED remains a training-free decoding strategy that does not require any additional fine-tuning or preference optimization, and thus introduces no extra training cost.
>
> **Table2: Comparison of different decoding strategies on POPE, MME, and MM-Vet benchmarks.**
>
> | **Method**  | **POPE \(R)** |           | **POPE \(P)** |           | **POPE \(A)** |           | **MME**        | **MMVet** |
> |-------------|--------------|-----------|--------------|-----------|--------------|-----------|----------------|-----------|
> |             | Accuracy ↑   | F1 Score ↑| Accuracy ↑   | F1 Score ↑| Accuracy ↑   | F1 Score ↑| Total Score ↑  | Overall ↑ |
> | RLAIF-V | 88.58        | 87.73     | 84.17        | 83.94     | 81.73        | 82.01     | 1768.42        | 25.3      |
> | ATED&   | 89.21        | **89.39** | 85.32 | 85.66  | 81.51        | 82.32 | 1780.69 | 25.0      |
> | ATED#   | **89.83**    | 89.35     | **86.71**    | **85.97**| **82.96**        | **82.78** | **1788.09**    | **26.6**  |
>
>
>
> [1] Yu T, Zhang H, Li Q, et al. Rlaif-v: Open-source ai feedback leads to super gpt-4v trustworthiness[C]//Proceedings of the Computer Vision and Pattern Recognition Conference. 2025: 19985-19995.

---

> ### Author Response · Authors · 2025-11-22
> **Response to Reviewer caNz（part 3/3）**
>
> > W3. While the paper reports improvements on hallucination-related metrics such as POPE, CHAIR, and MME (hallucination subset), it lacks qualitative and quantitative evaluation of overall generation quality (e.g., fluency, coherence, descriptive richness). For instance, benchmarks like RefoMB, LLaVA-Bench, MMStar, and MM-Vet could assess how ATED affects long-form captioning and reasoning under complex visual conditions. Without these results, it remains unclear whether ATED preserves or degrades naturalness in extended outputs.
> >
>
> &nbsp;
>
> **Answer:** We thank the reviewer for this insightful suggestion. As partially addressed in **Appendix E.4 (“Qualitative Analysis”)**, we already provide **a quantitative analysis on LLaVA-Bench**. Following your idea, we have further conducted experiments on **MM-Vet**, and the corresponding results are now reported in Tables 3 and 4 of this response.
>
> The results show that our ATED variants achieve the best performance on LLaVA-Bench, reaching the **metric Accuracy of 4.61** and improving over the strongest baseline by **6.2%**. On MM-Vet, our methods attain performanc better than RLAIF-V.
> &nbsp;
>
> **Table 3: Comparison with different decoding strategies on LLaVA-Bench.**
>
>
> | **Model** | **Method** | **Accuracy↑**        | **Detailedness↑**  |
> |-----------|------------|-----------------|--------------|
> | **LLaVA-1.5** | default    | 3.16            | 3.48         |
> |           | VCD        | 4.11            | 3.80         |
> |           | OPERA      | 4.25            | 3.89         |
> |           | SID        | 4.34            | **3.95**         |
> | **ATED\#**      |      -    | **4.61**            | 3.93         |
>
>
> **Table 4: Comparison with different decoding strategies on MM-Vet.**
> | **Model** | **Method**                  | **Overall ↑** |
> |-----------|-----------------------------|---------------|
> | LLaVA-1.5 | default                     |  25.2  |
> |           | VCD                         |  25.4    |
> |           | OPERA                       |   25.5   |
> |           | SID                         |   26.0             |
> |           | RLAIF-V                     |   $\underline{26.1}$  |
> | **ATED\#(W/O UGO)** |-    |    25.9              |
> |   **ATED\#**            |  -        |      **26.6**             |
>
> &nbsp;
>
> > W4. The paper focuses mainly on hallucination benchmarks but does not discuss whether ATED affects general-purpose multimodal understanding. Evaluations on textVQA, DocVQA, InfoVQA, and VQAv2 would help determine whether the ensemble decoding alters the model’s broader reasoning or comprehension abilities. It is important to verify that the hallucination reduction does not come at the cost of decreased general accuracy or robustness on standard multimodal benchmark.
> >
>
> &nbsp;
>
> **Answer:** We thank the reviewer for this valuable suggestion. Following the suggestion, we have added experiments on the **TextVQA** benchmark. To ensure a fair comparison, we replace the backbone used in ATED with a unified LLaVA-1.5-7B model based on Vicuna-v1.5-7B, and compare multiple decoding strategies on the **TextVQA val** split under this shared backbone (**see the Table 5 below**). As shown in the results, the proposed approach consistently achieves higher accuracy than the default decoding, whereas the preference-optimized model RLAIF-V exhibits a decrease of about 5.3%. These results indicate that, on standard multimodal benchmarks, **our method can maintain stronger robustness without sacrificing general accuracy**.
>
> &nbsp;
>
> **Table 5: Comparison of different decoding strategies on TextVQA.**
>
>
> | **Model**          | **Method** | **TextVQA val (Acc. ↑)** |
> |--------------------|------------|-----------------|
> | LLaVA-1.5 7B based | default    | 45.5            |
> |                    | VCD        | $\underline{47.3}$        |
> |                    | OPERA      | 46.6            |
> |                    | RLAIF-V    | 43.1            |
> |  Ensemble         | **ATED\#(W/O UGO)** |      46.7       |             |
> |           | **ATED\#**          |   **47.5**           |            |

---

### Author Response · Authors · 2025-12-04
**Response and Revision Summary for AC**

Dear Area Chair and Reviewers,

We sincerely thank the Area Chair for the time and consideration devoted to our paper ATED (Paper 8276). We greatly appreciate the constructive feedback and suggestions from all the reviewers. We are pleased that Reviewer caNz finds our adaptive uncertainty–based token-level ensemble decoding clear, well-motivated, and practically meaningful; Reviewer akFE recognizes ATED’s training-free, plug-and-play nature and strong effectiveness; Reviewer 5jsQ considers the topic interesting, the problem important, and the paper easy to follow; and Reviewer oHTM appreciates our use of a simple yet broadly applicable ensemble framework to mitigate object hallucination in LVLMs. We are especially grateful for the reviewers’ recognition of ATED’s broad applicability across multiple off-the-shelf LVLM backbones, as well as its strong empirical results and overall performance (Reviewers caNz, akFE, oHTM).

During the brief rebuttal period, only Reviewer oHTM responded, stating that most concerns had been addressed and that they would maintain their original positive score of **6**. We are pleased that our responses resolved this reviewer’s concerns.

We take the reviewers’ concerns seriously and address them as comprehensively as possible in our point-by-point response. Guided by these comments, we conducted new experiments, expanded analyses, and revised the manuscript, with all updates highlighted in blue. The key revisions and responses are as follows:

* **Inference latency:** We clarify the interpretation of Figure 4, conduct additional latency measurements, and report results in Table 4 (Sec. 3.7), resolving concerns raised by Reviewers 5jsQ and oHTM.
* **Experiments on RLAIF-V:** We add a systematic comparison with the training-based hallucination mitigation method RLAIF-V, as suggested by Reviewer akFE, and update Appendix E.3 with the corresponding detailed results.
* **More multimodal evaluation:** We add long-form generation and broader multimodal benchmarks (e.g., MMVet, TextVQA, MMMU) as suggested by Reviewers akFE and 5jsQ, and update Appendix E.4 with the corresponding results.
* **Comparison with ED:** We add new comparisons with the 2025 ED plug-and-play hallucination mitigator on the POPE and CHAIR benchmarks and update Appendix E.1 with the corresponding detailed results.
* **Ensemble baseline:** We clarify that Section 3.6 and Table 3 (row “Uniform”) in the original paper already present a multi-model ensemble baseline.
* **Alternative uncertainty metrics:** We clarify that Section 3.6 and Table 3 (row “Confidence-based”) already present a confidence-based weighting variant that uses maximum softmax probability as an alternative uncertainty signal.
* **Decoding pipeline:** We provide a more detailed description of the proposed decoding pipeline.
* **Model weight allocation:** We conduct a deeper analysis of model weight allocation during decoding on POPE and MME and update Appendix E.1 and E.2 with the corresponding detailed results.
* **Experiments on Qwen2.5-VL series models:** We apply the proposed ATED to the Qwen2.5-VL series models (Qwen2.5-VL-3B and Qwen2.5-VL-7B) and conduct experiments on the MMMU validation benchmark, as suggested by Reviewer 5jsQ.
* **Clarify the conceptual differences:** We clarify the conceptual differences between ATED and ED/VCD and provide an analysis of the contribution of UGO.
* **Statistical significance tests:** We report statistical significance tests and variance, as suggested by Reviewer oHTM.
* **Performance scaling with the ensemble size:** We further report how the performance of ATED scales with the ensemble size, as suggested by Reviewer oHTM.
* **Hyperparameter experiments:** We clarify the ablation studies on the noise step hyperparameter T and the hyperparameter α, which controls the strength of visual contrastive decoding.
* **Limitation of single-model heads:** We clarify that, under the current design, single-model internal heads or adapters cannot replace the proposed multi-model uncertainty weighting scheme.

We once again extend our sincere thanks to the Area Chair and all reviewers for their thoughtful feedback and constructive guidance.


Best, Authors of Paper 8276

---

### Meta-Review · Area_Chair_WTtd · 2025-12-26

**Summary:**

The paper proposes ATED, a training-free ensemble framework designed to mitigate multimodal hallucinations in large vision-language models (LVLMs). While the idea is clear and empirically effective on hallucination benchmarks, reviewers raise concerns about practicality, and limited evidence of broader impact, resulting in insufficient contribution strength.
Considering the reviewers’ concerns, we regret that the paper cannot be recommended for acceptance at this time. The authors are encouraged to consider the reviewers’ comments when revising the paper for submission elsewhere.

**Reviewer Concerns:**

Key concerns include (1) substantial inference cost and latency from multi-model decoding, (2) unclear real-world scalability, (3) incomplete comparisons with recent plug-and-play or training-based methods, (4) limited evaluation beyond hallucination benchmarks, and relatively weak novelty over prior ensemble or contrastive decoding approaches.

**Reviewer Scores:**

Reviewer scores are mixed. Soundness and presentation are generally rated good, but contribution is often fair. Ratings range from marginally below to marginally above acceptance, with several reviewers explicitly expressing openness to rejection, reflecting an overall borderline but insufficient consensus for acceptance.

---

### Decision · Program_Chairs · 2026-01-26

Reject